# The Bayesian Origin of the Probability Weighting Function in Human Representation of Probabilities

## Abstract

Understanding the representation of probability in the human mind has been of great interest to understanding human decision making. Classical paradoxes in decision making suggest that human perception distorts probability magnitudes. Previous accounts postulate a Probability Weighting Function that transforms perceived probabilities; however, its motivation has been debated. Recent work has sought to motivate this function in terms of noisy representations of probabilities in the human mind. Here, we present an account of the Probability Weighting Function grounded in rational inference over optimal decoding from noisy neural encoding of quantities. We show that our model accurately accounts for behavior in a lottery task and a dot counting task. It further accounts for adaptation to a bimodal short-term prior. Taken together, our results provide a unifying account grounding the human representation of probability in rational inference.

## 1 Introduction

It is a long-standing observation that human representation of probability is distorted. In decision-making under risk, this manifests as systematic deviations from the Expected Utility framework, as highlighted by the Allais paradox (Allais, 1953). Prospect Theory (Kahneman & Tversky, 1979) addressed these deviations by introducing a *probability weighting function*, typically inverse S-shaped: small probabilities are overweighted, while large probabilities are underweighted (Figure 1A). This function has been central in explaining a wide range of behavioral anomalies (Ruggeri et al., 2020).

However, a fundamental question remains unanswered: *what is the origin of the probability weighting function?* Classical approaches have proposed parametric forms (e.g. Prelec, 1998; Zhang & Maloney, 2012), which describe but do not explain its shape. More recent work (e.g. Fennell & Baddeley, 2012; Zhang et al., 2020; Khaw et al., 2021; Frydman & Jin, 2023; Bedi et al., 2025; Enke & Graeber, 2023) suggests that, rather than following a deterministic transformation, probabilities are imprecisely encoded in the mind, and that properties of this encoding give rise to distortions, for instance because humans combine noisy encodings with prior expectations about the value of probabilities. However, the specific process by which the S-shaped distortion results remains unclear, with ideas such as log-odds-based transformations (Zhang et al., 2020; Khaw et al., 2021), biases away from the bounds of the response range (Fennell & Baddeley, 2012; Bedi et al., 2025), or efficient coding (Frydman & Jin, 2023).

In this paper, we show that probability representation can be parsimoniously explained in terms of **optimal decoding from noisy internal representations** in the brain (Figure 1B). In our Bayesian framework, probabilities are imprecisely encoded and decoded via Bayes risk minimization, naturally giving rise to systematic distortions in perceived probability. This account not only unifies prior proposals as special cases but also yields testable theoretical predictions. Specifically, our contributions are two-fold:

1. We theoretically disentangle the predictions made by distinct modeling proposals. By deriving analytical decomposition of bias, we separate the contributions of prior attraction, encoding-induced repulsion and boundary regression, and clarify that these modeling proposals rely on different resource allocations (Section 3).

2. We conduct the first broad empirical evaluation comparing these models against data from three empirical paradigms. Unlike prior work that typically focuses on a single task, we compare five model classes across judgment of relative frequency, pricing, and choice tasks, showing the specific limitations of each account (Section 4).

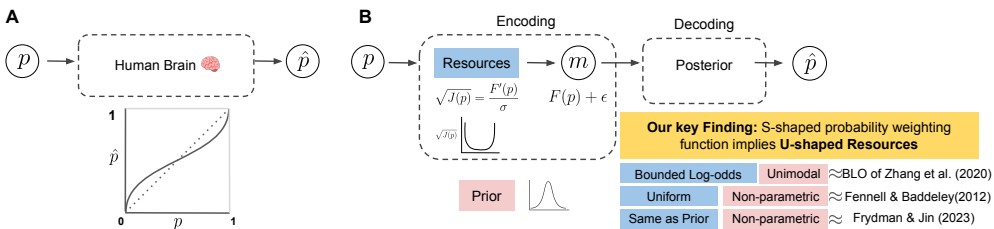

Figure 1: Distorted probability perception and our Bayesian account. (A) Human decisions systematically distort probabilities, producing the inverse S-shaped probability weighting function central to Prospect Theory. (B) We study a Bayesian encoding–decoding framework: true probabilities $p$ are encoded noisily, combined with a prior, and decoded into perceived probabilities $\hat{p}$. Encoding resources allocated to a probability $p$ are proportional to the slope of the encoding function $F$ and $\sigma$ is the sensory noise standard deviation. This framework explains the origin of the weighting function and predicts that the observed S-shape implies a U-shaped allocation of encoding resources.

## 2 BACKGROUND AND RELEVANT WORK

A large body of work in economics and psychology models probability distortion with parametric forms of the **probability weighting function**. Classical examples include the inverse-S shaped forms in Prospect Theory (Tversky & Kahneman, 1992), the Prelec function (Prelec, 1998), and the Linear-in-Log-Odds (LILO) model (Zhang & Maloney, 2012), which assumes that under the log-odds transform, the perceived probability $\hat{p}$ is linear in the true probability $p$ ($\gamma, \beta$ are free parameters):

$$\hat{p} = \lambda^{-1}(\gamma\lambda(p) + \beta), \quad \lambda(p) := \log\frac{p}{1-p}. \tag{1}$$

These models successfully capture behavioral regularities such as overweighting of small probabilities and underweighting of large ones. However, they remain primarily descriptive.

More recent work has shifted focus from deterministic transformations to the idea that probabilities are encoded imprecisely in the brain. Under this view, distortions arise as systematic consequences of noisy internal representations. Several mechanisms have been proposed. **Regression-based accounts** emphasize biases away from the boundaries of the probability scale (Fennell & Baddeley, 2012; Bedi et al., 2025). **Log-odds based approaches** assume that internal representations of probability are approximately linear in log-odds space (Khaw et al., 2021; 2022). Other recent work highlights **efficient-coding accounts** (Frydman & Jin, 2023) and optimal inference under noisy encodings (Juechems et al., 2021; Enke & Graeber, 2023). These models have generally not been broadly evaluated against behavioral data, and their differing predictions are not understood either theoretically or empirically. One related model has already been evaluated on several behavioral datasets and shown success in outperforming fixed probability weighting functions, the **Bounded Log-Odds (BLO) model** (Zhang et al., 2020). BLO assumes that probability $p$ is first mapped to log-odds and truncated to a bounded interval $[\Delta^-, \Delta^+]$( 2). The clipped value is then linearly mapped to an encoding $\Lambda(p)$ on an interval $[-\Psi, \Psi]$( 3), and then combined with an "anchor" $\Lambda_0$, where $\kappa > 0$ is a free parameter( 4). This encoding is subject to Gaussian noise and then decoded into an estimate of the probability by applying the inverse log-odds function( 5).

$$\lambda(p) = \log\frac{p}{1-p}, \qquad \Gamma(\lambda) = \min\left(\max(\lambda, \Delta^-), \Delta^+\right) \tag{2}$$

$$\Lambda(p) = \frac{\Psi}{(\Delta_+ - \Delta_-)/2}\left(\Gamma(\lambda(p)) - \frac{\Delta_- + \Delta_+}{2}\right) \tag{3}$$

$$\hat{\Lambda}_\omega(p) = \omega_p \cdot \Lambda(p) + (1 - \omega_p) \cdot \Lambda_0, \qquad \omega_p = \frac{1}{1+\kappa V(p)}, \qquad V(p) \propto p(1-p), \tag{4}$$

$$\hat{\pi}(p) = \lambda^{-1}(\hat{\Lambda}_\omega(p) + \epsilon_\lambda), \quad \epsilon_\lambda \sim \mathcal{N}(0, \sigma_\lambda^2). \tag{5}$$

## 3 A GENERAL BAYESIAN FRAMEWORK FOR PERCEIVED PROBABILITY

The traditional approach assumes that probabilities are deterministically transformed, using a domain-specific, nonlinear function (Tversky & Kahneman, 1992). Here, we propose a different account based on general principles of perceptual processing. Specifically, we model the probability $p$ as encoded into a noisy internal signal $m$, from which the decision maker derives an estimate $\hat{p}$ by minimizing Bayes risk. Distortions of probabilities arise because the optimal Bayesian estimate is typically biased (e.g. Knill & Richards, 1996; Weiss et al., 2002; Körding & Wolpert, 2004). This encoding-decoding approach has been successful in accounting for biases in the perception of numerosity, orientation, color, time intervals, and subjective value (e.g. Polanía et al., 2019; Summerfield & De Lange, 2014; Fritsche et al., 2020; Woodford, 2020; Jazayeri & Shadlen, 2010; Girshick et al., 2011; Wei & Stocker, 2015; 2017; Hahn & Wei, 2024).

Formally, the stimulus $p$ is mapped to an internal, noisy sensory measurement $m$ representing the neural encoding in the brain, which can be abstracted as a one-dimensional transformation:

$$m = F(p) + \epsilon, \quad \text{where } \epsilon \sim \mathcal{N}(0, \sigma^2) \tag{6}$$

where $F : [0,1] \to \overline{\mathbb{R}}$ is a general, strictly monotone increasing and smooth function. $m$ represents the neural population code for $p$ in the brain. The slope of $F(p)$ is the encoding precision, with Fisher Information (FI) $\mathcal{J}(p) = (F'(p))^2/\sigma^2$, i.e., greater slope of $F$ indicates greater encoding precision. Following Hahn & Wei (2024), we refer to $\sqrt{\mathcal{J}(p)}$ as the *(encoding) resources* allocated to $p$. Given $m$ and a prior distribution $P_{prior}(p)$, Bayesian inference yields a posterior $P(p|m)$. We assume that a point estimate $\hat{p}(m)$ is derived as the posterior mean, which minimizes the Bayes risk for the mean-squared loss function. The model's behavior is therefore governed by the encoding $F$ and the prior $P_{\text{prior}}$, both of which we will infer from behavioral data.

Multiple existing accounts represent special cases of this framework (Fennell & Baddeley, 2012; Frydman & Jin, 2022; Bedi et al., 2025; Khaw et al., 2021; 2022; Enke & Graeber, 2023), such as with a uniform FI (Fennell & Baddeley, 2012), a log-odds-based encoding (Khaw et al., 2021; 2022), or efficient coding (Frydman & Jin, 2022). None of these have been evaluated broadly against behavioral data, and their differing predictions are not understood theoretically or empirically. We will discuss these accounts in Section 3.1.2.

### 3.1 THEORETICAL RESULTS

#### 3.1.1 THE PREDICTIONS OF OUR FRAMEWORK

We now analytically derive testable predictions from the Bayesian framework. Unlike traditional deterministic probability weighting functions, the Bayesian account treats the perceived probability as stochastic, depending on the noisy encoding $m$. The distortion of probability entailed by the Bayesian model can be studied by its bias, i.e., the average deviation of the estimate from the true probability across trials:

$$\text{Bias}(p) := \mathbb{E}[\hat{p}|p] - p \tag{7}$$

Whereas prior work on the functional form of the bias of Bayesian models of perception largely focused on unbounded or circular stimulus spaces, or stimuli far away from a boundary (Hahn & Wei, 2024; Wei & Stocker, 2017; Prat-Carrabin & Woodford, 2021; Morais & Pillow, 2018; Stocker & Simoncelli, 2006), the bounded nature of probabilities requires a treatment fully accounting for the boundary (see Appendix B.1.1 for the proof):

**Theorem 1.** *There are functions $A_{1,\sigma}, A_{2,\sigma}, A_{3,\sigma} : [0,1] \to \mathbb{R}_+$ that are bounded, positive, and satisfy*

$$A_{...,\sigma}(p) \leq \phi\left(\frac{\sigma}{\min\{|F(p) - F(0)|, |F(p) - F(1)|\}}\right) \tag{8}$$

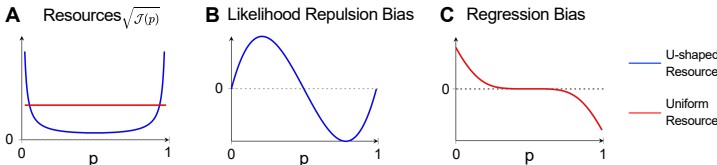

Figure 2: Impact of encoding on bias. (A) A U-shaped (blue) and uniform (red) encoding allocation resources $\sqrt{\mathcal{J}(p)}$. (B) The U-shaped encoding generates an S-shaped Likelihood Repulsion bias, $\propto \frac{\mathrm{d}}{\mathrm{d}\,p}(1/\mathcal{J}(p))$. (C) The uniform encoding generates a Regression bias $\propto A_{1,\sigma}(p) \cdot \frac{\mathrm{sign}(0.5-p)}{\sqrt{\mathcal{J}(p)}}$, maximized at 0,1. Both biases generate overestimation of small and underestimation of large probabilities, but with distinct shapes.

*for some nondecreasing function $\phi : \mathbb{R} \to \mathbb{R}$ with $\lim_{x\downarrow 0}\phi(x) = 0$, across all $p \in (0,1); \sigma > 0$. Then, at any $p \in (0,1)$, the Bayesian model has the bias, as $\sigma \to 0$:*

$$\mathrm{Bias}(p) = \underbrace{A_{1,\sigma}(p) \cdot \frac{\mathrm{sign}(0.5-p)}{\sqrt{\mathcal{J}(p)}}}_{\text{Regression from Boundary}} + \underbrace{(1 - A_{2,\sigma}(p)) \cdot \frac{\mathrm{d}}{\mathrm{d}\,p}\left(\frac{1}{\mathcal{J}(p)}\right)}_{\text{Likelihood Repulsion}}$$

$$+ \underbrace{(1 - A_{3,\sigma}(p)) \cdot \frac{1}{\mathcal{J}(p)} \cdot \frac{\mathrm{d}}{\mathrm{d}\,p}\left(\log P_{prior}(p)\right)}_{\text{Prior Attraction}} + \mathcal{O}(\sigma^4)$$

The bias arises from three sources, using the terminology of Hahn & Wei (2024). The first term is **regression** effect towards $p = 0.5$ (first term); a **repulsion** away from high-FI stimuli (second term), and **attraction** towards regions of high prior density (third term). All three components are modulated by terms $A_{1,\sigma}, A_{2,\sigma}, A_{3,\sigma}$ that intuitively represent the effect of the boundary: Eq. 8 expresses that these terms vanish when the noise magnitude $\sigma$ is small in relation to the distance of $F(p)$ from both $F(0)$ and $F(1)$: that is, how close $p$ is to one of the two boundaries in the neural encoding space. In particular, if $F(0) = -\infty, F(1) = \infty$, the $A_{\dots,\sigma}$ factors vanish everywhere. On the other hand, when $p$ is close to a boundary, these factors can substantially affect the bias.

We now explain how this theorem drives key predictions. We first consider the first two terms, which are independent of the prior. The probability weighting function $w(p)$ is typically taken to be S-shaped and to have fixed points at the boundaries, i.e., $w(0) = 0$ and $w(1) = 1$ (Figure 1A) (Prelec, 1998; Tversky & Kahneman, 1992). For the bias to vanish at the boundaries, the Regression term (Figure 2C) must vanish, which requires $\mathcal{J}(0), \mathcal{J}(1)$ approaching infinity. As infinite precision is biologically implausible in neural populations, we predict a very high but finite $\mathcal{J}(0), \mathcal{J}(1)$, leading to near-zero regression bias at the endpoints. If Regression is neutralized, the Likelihood Repulsion term $\propto \frac{\mathrm{d}}{\mathrm{d}\,p}(1/\mathcal{J}(p))$ remains; it is S-shaped (positive for small $p$, negative for large $p$, Figure 2B) if and only if $\mathcal{J}(p)$ is U-shaped (Figure 2A). We overall thus predict U-shaped Resources:

Prediction 1: Standard probability weighting functions imply U-shaped resources with peaks at 0 and 1.

The Attraction component allows general biases depending on the shape of the prior. Under the Bayesian framework, this is predicted to depend on the stimulus distribution: a change to the prior – for instance, due to exposure to a new stimulus distribution – would lead to a change in the attraction term, and potentially a deviation from the S-shaped bias:

Prediction 2: The Prior Attraction term provides a mechanism to produce deviations from the S-shaped probability weighting function.

Besides the bias, another key signature of behavior is the response variability, $SD(\hat{p}|p)$. Our analysis shows that it is shaped by both FI and prior (Theorem 5 in Appendix B.1.1). A change in the prior, for instance, should therefore lead to a predictable change in the response variability.

> Prediction 3: The prior distribution impacts the response variability.

Finally, we consider the overall performance of the estimator by analyzing its Mean Squared Error (MSE). The Bayesian estimate is designed to minimize the MSE, and, when noise is low, achieves the Cramer-Rao Bound, $MSE = \frac{1}{\mathcal{J}(p)} + O(\sigma^4)$ ( Theorem 6 in Appendix B.1.1).

> Prediction 4: Bayesian model predicts optimal MSE, with zero estimation error as internal noise diminishes.

We will test these theoretical predictions using empirical results in Section 4.

### 3.1.2 DIVERGENT PREDICTIONS FROM ALTERNATIVE MODELS

We next contrast Predictions 1–4 with the implications of alternative models.

**Prediction 1** is shared with models that assume a log-odds encoding (Zhang et al., 2020; Khaw et al., 2021; 2022), other encodings with U-shaped FI (Enke & Graeber, 2023), or efficient coding for a U-shaped prior (Frydman & Jin, 2023). In this case, the nonuniformity of the encoding makes the Repulsion term (Figure 2B) central. In contrast, Fennell & Baddeley (2012) and Bedi et al. (2025) attribute the S-shaped probability weighting entirely to the boundary-induced regression (Figure 2C), potentially modulated by the Prior Attraction. The Regression effect is strongest at 0 and 1 (unless $\mathcal{J}(p)$ approaches infinity there, in which case it vanishes), i.e., such accounts predict the distortion to be *particularly large* at the boundaries (Figure 2C), which conflicts the S-shape traditionally assumed. Frydman & Jin (2023) propose that the encoding is optimized for mutual information under a prior, which leads to $\sqrt{\mathcal{J}(p)} \propto p_{prior}(p)$ (see Appendix B.2). They note that, if the prior is U-shaped, the resulting model would produce an S-shaped bias. However, they do not explicitly recover the encoding from the behavioral data; we'll be able to do this on several datasets, and find that prior and encoding are not always matched in behavioral data.

**Prediction 2** is shared with models incorporating Bayesian decoding (Fennell & Baddeley, 2012; Bedi et al., 2025); the BLO model allows attraction to the anchor $\Lambda_0$ but no general priors. In contrast, if prior and encoding are matched by efficient coding as $\sqrt{\mathcal{J}(p)} \propto p_{prior}(p)$, the posterior mean is biased *away* from the prior mode (Wei & Stocker, 2015).

**Prediction 3** is a general prediction of Bayesian decoding; in contrast, in BLO, response variance is mainly determined by $\Delta^-, \Delta^+$ and $\sigma$, but not the anchor $\Phi_0$ – diverging from the Bayesian model (Appendix, Theorem 10).

**Prediction 4** about the optimality is again shared by models assuming Bayesian decoding. BLO differs here as well: even when $\sigma, \kappa = 0$, the MSE of its decoded estimate is nonzero everywhere in $(0, 1)$, entailing suboptimal estimation.

### 3.2 BAYESIAN MODELS WITH LOG-ODDS ENCODING

The Bayesian model can accommodate a general mapping $F$ and we will be able to infer it from data. Here, we justify the popular choice of a log-odds mapping (as assumed by Khaw et al. (2021; 2022); Zhang et al. (2020)), and how this leads to a reinterpretation of BLO as an approximation of the Bayesian model. Suppose that observers encode positive counts ($\kappa_+$) and negative counts ($\kappa_-$):

$$m = \begin{pmatrix} F_{num}(\kappa_+) + \epsilon_1 \\ F_{num}(\kappa_-) + \epsilon_2 \end{pmatrix} \qquad \epsilon_1, \epsilon_2 \sim \mathcal{N}(0, \sigma^2) \tag{9}$$

where $p = \frac{\kappa_+}{\kappa_+ + \kappa_-}$. Based on research on magnitude perception, we take $F_{num}$ to be consistent with Weber's law by assuming the form (e.g. Petzschner & Glasauer, 2011; Dehaene, 2003):

$$F_{num}(\kappa) = \log(\kappa + \alpha) \qquad \text{hence} \qquad m = \begin{pmatrix} \log(pN + \alpha) + \epsilon_1 \\ \log((1 - p)N + \alpha) + \epsilon_2 \end{pmatrix}$$

where $\alpha > 0$ prevents an infinite FI at zero (Petzschner & Glasauer, 2011), and $N = \kappa_+ + \kappa_-$. What is an optimal 1D encoding $m_{1D} \in \mathbb{R}$ of the rate $p = \frac{\kappa_+}{\kappa_+ + \kappa_-}$? We focus on linear encodings with coefficients $w_1, w_2$:

$$m_{1D} := w_1(\log(pN + \alpha) + \epsilon_1) + w_2(\log((1 - p)N + \alpha) + \epsilon_2) \tag{10}$$

This uniquely encodes $p$ only if $w_1 w_2 < 0$ because of monotonicity. Symmetry thus suggests $w_1 = -w_2$, equivalent to a log-odds encoding smoothed at the boundaries ($\beta > 0$):

$$F(p) := \log \frac{p + \beta}{(1 - p) + \beta} \qquad \sqrt{\mathcal{J}(p)} = \frac{1}{\sigma} \cdot \left( \frac{1}{p + \beta} + \frac{1}{(1 - p) + \beta} \right) \qquad (11)$$

where, for fixed $N$, we absorbed $N$ into $\beta = \alpha/N$. Note that this $\mathcal{J}(p)$ is U-shaped as in Figure 2A. This form for $F$ allows us to interpret BLO as an approximation to the Bayesian model. First, in the limit where $\beta \to 0$ (i.e., unbounded log-odds encoding), the Bayesian bias comes out to (see Corollary 19 in Appendix B.1.1):

$$\mathbb{E}[\hat{p}|p] - p = \underbrace{\sigma^2 p^2 (1 - p)^2 \left( \log P_{prior}(p) \right)'}_{\text{Attraction}} + \underbrace{2\sigma^2 p(1 - p)(1 - 2p)}_{\text{Repulsion}} + \mathcal{O}(\sigma^4) \qquad (12)$$

The first term depends on the prior distribution. As expected, the second term describes an S-shaped bias (Figure 2B). Indeed, when noise parameters are small, the BLO model matches the bias of this Bayesian model with a specific unimodal prior (Proof in Appendix B.1.2):

**Theorem 2.** *The BLO model in the limit of untruncated log-odds ($\Delta_+ \to \infty, \Delta_- \to -\infty$), and a Bayesian model with unbounded log-odds encoding ($\beta = 0$) and a specific unimodal prior $P_{prior}(p)$ (depending on $\Lambda_0$) have the same bias up to difference $\mathcal{O}(\kappa^2 + \sigma^4)$.*

This result suggests that one can reinterpret BLO as an approximation of a Bayesian model with log-odds encoding and a unimodal prior, albeit with divergences in the response distribution due to its sub-optimal decoding process. Our empirical results detailed below support this view, showing that the Bayesian model consistently provides better fit to behavior compared to the BLO model.

## 4 EMPIRICAL VALIDATION OF THE BAYESIAN FRAMEWORK

We empirically evaluate our theory within a unified Bayesian framework. To test our model and predictions against the full landscape of existing theories, we perform a head-to-head comparison of our Bayesian account (U-shaped FI with task-dependent priors) against: i) Fixed probability distortion function, ii) BLO model (Zhang et al., 2020), iii) Bayesian model with uniform encoding (Fennell & Baddeley, 2012; Bedi et al., 2025), and iv) Efficient coding with matched prior and encoding (Frydman & Jin, 2023). This comparison allows us to pinpoint the specific limitations of each account.

First, we validate Predictions 1, 3, 4 using a *judgment of relative frequency* (JRF) task, where subjects estimate the proportion of dots. Here, we show that uniform encoding models cannot capture the S-shaped bias patterns, while BLO cannot account for response variability. Second, we assess the generality of our framework by applying it to a different domain, *decision-making under risk*, showing that our account outperforms classical fixed distortion functions and other accounts. Third, we test our model's ability to adapt to different stimulus statistics to test Prediction 2 and demonstrate that BLO cannot explain human adaptation to bimodal distributions.

### 4.1 VALIDATING PREDICTIONS ON JUDGMENT OF RELATIVE FREQUENCY (JRF) DATA

We analyze data from Zhang et al. (2020) and Zhang & Maloney (2012), where subjects ($N = 86$) judge the percentage of dots of a target color in arrays of black and white dots on a gray background.[1]

Given that BLO outperforms traditional fixed distortion functions (Zhang et al., 2020), we treat it as the strong baseline for parametric models. We further investigate whether our Bayesian account offers a more unified explanation than competing accounts of uniform encoding and efficient coding. We evaluated Bayesian variants combining Gaussian prior with fitted mean and variance (GaussianP) or nonparametric prior(FreeP) with three encodings: uniform encoding, $F(p) = p$ (UniformE), bounded-log-odds encoding (BoundedLOE)[2], and freely fitted $F$ (FreeE). We also included a matched prior–encoding model to test the efficient coding account. In both BLO and the

---

[1]A complete description of the tasks and datasets used in this paper is available in Appendix C.2.

[2]Formulated as $F(p) = \log(p + \beta_1) - \log(1 - p + \beta_2)$, similar to 11 but with separate upper and lower bounds, in line with BLO's $\Delta_-, \Delta_+$ bounds.

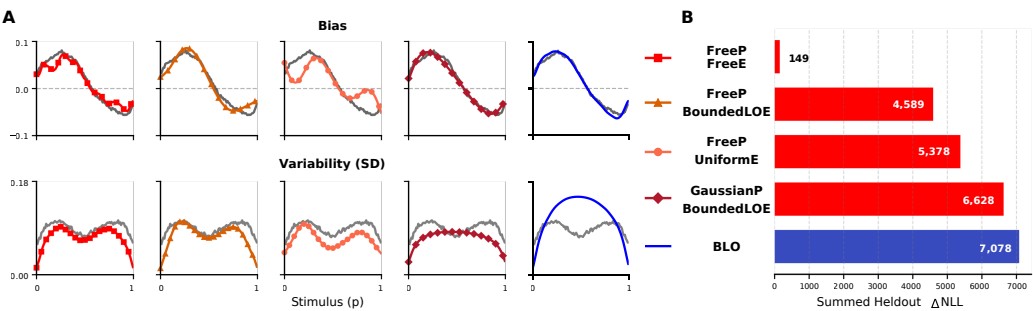

Figure 3: Results for JRF (Section 4.1). Each Bayesian model variant is defined by a prior and encoding; see text for the shorthands. (A) Model fits to response bias (top) and variability (bottom). Gray curves show human data. The S-shaped bias is only explained by non-uniform encodings (BoundedLOE, FreeE); the variability is only captured by the nonparametric prior (FreeP). Appendix Figures 10–15 provide per-subject results for resources, bias, and variability. (B) Model fit on held-out data (Summed Heldout ΔNLL; lower is better). All bayesian models outperform BLO.

Bayesian model, parameters are optimized for each subject individually by maximizing trial-by-trial data likelihood. We fitted the Bayesian model using the method of Hahn & Wei (2024), which permits identifying the encoding nonparametrically from behavioral responses (Hahn et al., 2025).[3]

We evaluate model fit using negative log-likelihood (NLL) on held-out data, and compare models using the Summed Held-out ΔNLL, which aggregates across subjects the difference between each model's held-out NLL and that subject's best model. As shown in Figure 3B, all Bayesian models achieve better fit than BLO, including a Gaussian prior with Bounded Log-Odds encoding at a similar number of parameters. Allowing most flexibility in the prior or encoding(FreeP, FreeE) further improves performance. We also find that the model with matched prior and encoding achieved an inferior fit compared to the freely fitted model (Appendix Figure 8A).

First, we examine the shape of resources (**Prediction 1**). Models with nonuniform encoding capture the characteristic S-shaped bias in behavior, whereas the variant with uniform encoding fails to do so (Figure 3A, top). To pinpoint the mechanism, we decomposed the bias (Figure 4). The UniformE model exhibits zero Likelihood Repulsion, forcing it to rely exclusively on Boundary Regression and Prior Attraction, which are insufficient to recover the full S-shaped bias. In contrast, the recovered nonuniform resources are consistently U-shaped with peaks near the boundaries (Figure 4), a pattern that generates the required repulsion. While the BLO model also shares the same overall U-shaped resources, it has two discontinuities, a property proved in Appendix Theorem 8 and illustrated in Appendix Figure 11.Furthermore, contrary to the efficient encoding, which predicts that encoding matches the prior, we found that the recovered resources and prior have distinct shapes (Appendix Figure 32). Consequently, enforcing a matched prior-encoding constraint yields a substantially inferior fit compared to decoupled fitting of both components (Appendix Figure 8).

Next, we test the fit of response variability (**Prediction 3**). The bimodal pattern of response variability with a dip near 0.5 is only explained by models with a nonparametric prior (FreeP) (Figure 3A, bottom). Such priors develop a sharp peak around 0.5 (Figure 12)which mathematically suppresses variance. In contrast, The BLO model predicts variability based on noise in log-odds space, typically implying a simple inverted-U shape (Theorem 10), failing to capture this structure.

Finally, for **Prediction 4**, the close match of the Bayesian best-fitting model to both bias and variability suggests that human behavior is consistent with optimal decoding, whereas the BLO model substantially overestimates this variability and thus the mean square error of the data. Taken together, these empirical results validate prediction 1, 3 and 4, showing that probability distortion stems from optimal decoding of U-shaped neural representations rather than from uniform encoding or deterministic transformations as in BLO.

---

[3]Across all datasets, we ran all models using a grid of 200 points and regularization strength 1.0, ensuring a head-to-head comparison.

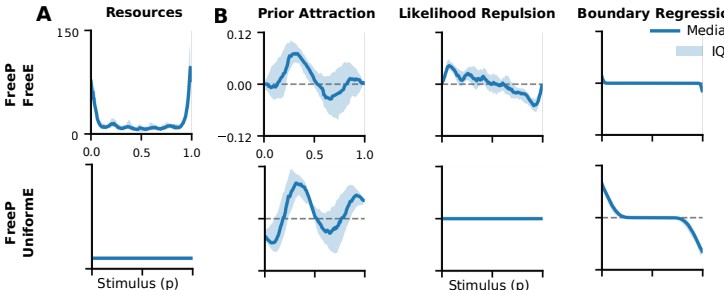

Figure 4: Resources (A) and bias decomposition (B) of FreeP FreeE and FreeP UniformE models for JRF data. IQR denotes interquartile range across subjects. In the best-fitting FreeP FreeE model, boundary regression plays only a small role, whereas Likelihood Repulsion consistently produces a bias away from 0 and 1.

## 4.2 TESTING THE GENERALITY ON DECISION-MAKING UNDER RISK

We next examine whether the principle of optimal decoding that explained probability distortion in perception also extends to economic decision-making under risk. To this end, we analyze two datasets where subjects evaluated two-outcome gambles $(x_1, p; , x_2, 1-p)$ with real-valued payoffs. The tasks share the same basic format of binary gambles, though their procedures differ slightly, and both are analyzed within a common modeling framework.

All models are formulated within the standard Cumulative Prospect Theory (CPT) framework (Tversky & Kahneman, 1992), in which the subjective utility of a gamble combines a value function $v(x)$ with a probability weighting function $w(p)$. For pure gains,

$$\text{Utility} := v(x_1) \cdot w(p) + v(x_2) \cdot (1 - w(p)) \quad v(x) = |x|^\alpha, \tag{13}$$

while mixed gambles employ a two-part value function for loss aversion and separate weighting functions for gains and losses ($w^+(p), w^-(p)$; see Appendix Eq. 39).

The key distinction across models lies in the form of the probability weighting function $w(p)$. As in the JRF task, our central hypothesis is that $w(p)$ is not an arbitrary parametric form but arises from the same optimal decoding mechanism that accounts for perceptual probability distortion. This implies that, in decision-making, our model should outperform competing parametric accounts and that the S-shaped weighting function emerges from a non-uniform encoding (**Prediction 1**).

**Pricing data** We first test our framework on the pricing data from Zhang et al. (2020). On each trial, subjects ($N = 75$) chose between a two-outcome monetary gamble and a sure amount, with the sequence of choices adaptively converging to the Certainty Equivalent (CE) of the gamble.

Given that Zhang et al. (2020) showed BLO as a superior account compared to traditional functions in this pricing task as well, it serves as the primary benchmark of parametric models. All models are expressed within the CPT framework (Eq. 13), differing only in the form of the probability weighting function $w(p)$. BLO treats $w(p)$ as a deterministic transformation of the objective probability, whereas in our Bayesian framework, $w(p)$ arises from stochastic encoding and optimal decoding. Within the Bayesian class, we test uniform, bounded log-odds, and freely fitted encodings, allowing direct comparison both with BLO and across Bayesian variants. Directly fitting CE responses does not isolate the contribution of $w(p)$, since each CE reflects both the utility function and probability weighting. Because BLO defines $\hat{\pi}(p)$ deterministically, it can be directly fit to CE responses, as in Zhang et al. (2020). By contrast, in the Bayesian framework, $\hat{\pi}(p)$ is derived from a stochastic encoding. For clearer inference on $w(p)$, we include an intermediate step: each reported CE is first inverted through the CPT utility function to obtain an implied probability weight for that trial, and then the Bayesian encoding parameters are estimated from these trial-level weights. With these encoding parameters fixed, the model predicts a distribution over CEs for each gamble; we then re-optimize the utility exponent $\alpha$ and CE noise variance $\sigma_{\text{CE}}^2$ to maximize the likelihood of the observed CE data. This ensures a fair evaluation that the final model comparison is based on the likelihood of the original CE responses for all models.

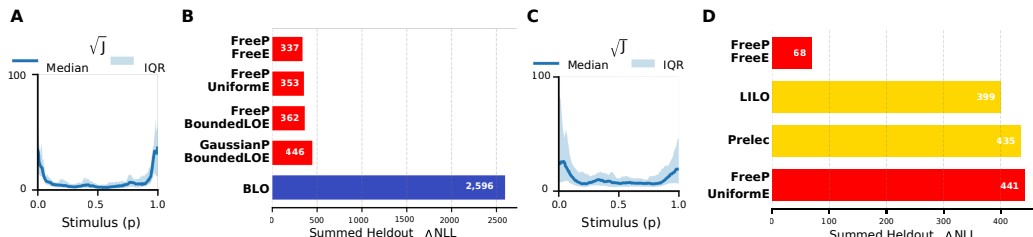

Figure 5: Across both pricing (A–B) and choice tasks (C–D), Bayesian models recover U-shaped resources with peaks near 0 and 1, consistent with **Prediction 1**. In the pricing task, all Bayesian variants outperform BLO, though the fit do not clearly distinguish between U-shaped and uniform encoding (B). In the choice task, however, the freely fitted encoding provides a markedly better fit than a uniform encoding and also outperforms standard parametric weighting functions (C–D).

As shown in Figure 5B, all Bayesian variants provide a markedly better fit than BLO. Moreover, our Bayesian framework inherently predicts trial-to-trial variability in responses with its stochastic encoding, a feature that deterministic models like BLO cannot capture. Within the Bayesian class, the freely fitted model recovers the predicted U-shaped resources (**Prediction 1**; Figure 5A). However, its quantitative fit is only marginally better than that of the uniform encoding model. This ambiguity motivates our next analysis, where we turn to a choice dataset to seek more conclusive evidence.

**Choice Data** To further test the generality of our framework, we analyzed a subset of the Choice Prediction Competition 2015 dataset (Erev et al., 2017). Specifically, we focused on the subset of two-outcome gambles with fully described probabilities and uncorrelated payoffs (Amb=0, Corr=0, LotNum=1), yielding 187,150 trials across 153 subjects.

In the choice task, probabilities of gains and losses are modeled with separate weighting functions $w^+(p)$ and $w^-(p)$, following CPT. The Bayesian model implements these by fitting separate priors for gains and losses while sharing the same encoding, with $w^\pm(p)$ computed as the expectation of $\hat{p}$ under the encoding distribution. As parametric benchmarks, we included the LILO function (Eq. 1) and the Prelec function (Prelec, 1998). BLO was not considered here, as its original specification does not extend to choice tasks. Because the observed responses are binary choices rather than CE values, we modeled choice probabilities with a logit rule applied to the difference in subjective utilities, with a temperature parameter $\tau$. All parameters were estimated separately for each subject.

As shown in Figure 5D, the Bayesian model with freely fitted prior and encoding fits the data better than both Prelec and LILO, demonstrating the framework's generality to the choice task. The fitted resources are U-shaped, with high sensitivity near 0 and 1 (Figure 5C). Crucially, the freely fitted encoding performs significantly better than a model constrained to a uniform encoding. This finding confirms **Prediction 1** and provides strong evidence that optimal decoding from U-shaped resources generalizes to decision-making under risk. In sum, these findings extend the validity of Prediction 1 to the economic domain, showing that U-shaped resources are essential for capturing the discriminability required in risky choice, whereas uniform encoding is structurally insufficient.

### 4.3 TESTING BAYESIAN ACCOUNT VIA ADAPTATION TO STIMULUS STATISTICS

A central prediction of Bayesian accounts is that priors adapt to the statistics of the environment. In our setting, this implies that a bimodal stimulus distribution should induce a bimodal prior, yielding biases with attraction points near each mode (**Prediction 2**). BLO, by construction, is limited to a single anchor and therefore can only predict one point of attraction. Efficient coding with matched prior and encoding (Frydman & Jin, 2023) predicts that resources reallocate to match the bimodal prior, which would theoretically generate a repulsion bias *away from* the modes (Wei & Stocker, 2015). In sharp contrast, our account predicts attraction *towards* the modes of the prior.

To test these predictions, we conducted an experiment using the same dot-counting task as in Section 4.1[4], but replaced the uniform stimulus distribution with a bimodal distribution of dot propor-

---

[4]The experimental design was approved by the Ethical Review Board of (ANONYMIZED).

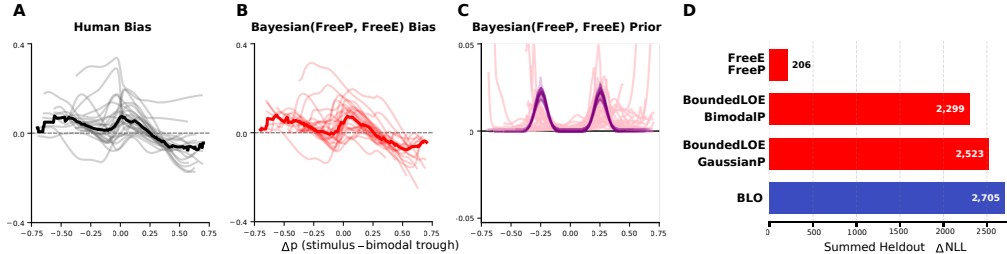

Figure 6: Adaptation to bimodal stimulus statistics.(A) Human bias, aligned so the trough between stimulus peaks is at 0 ($\Delta p$ = stimulus $p$ minus trough). Thin curves are individual subjects; solid is median. (B) Bias predicted by the Bayesian model with freely fitted prior and encoding (FreeP, FreeE). (C) Priors recovered by FreeP, FreeE. (Pink: subjects; Purple: ground truth) (D) Model fit (Summed Heldout $\Delta$NLL; lower is better). Models with flexible prior and encoding (FreeP, FreeE) significantly outperform constrained accounts (PriorMatchedE) and BLO.

tions, with two peaks equidistant from 0.5. 26 subjects completed the task. Within the Bayesian framework, we extended the set of priors to include a mixture-of-two-Gaussians prior (BimodalP) to explicitly test adaptation, and a matched prior–encoding model to test the efficient coding account.

The bimodal stimulus distribution induced a distinctive bias pattern in human responses, with two cross-over points centered on the distribution's modes (Figure 6A). Bayesian models reproduced this pattern: the FreeP+FreeE variant not only captured the observed bias (B) but also recovered a bimodal prior with peaks aligned to the stimulus modes (C). Quantitatively, Bayesian models outperformed the matched prior-encoding model(efficient coding account) and BLO (D), with BimodalP providing a better fit than a unimodal Gaussian prior, and FreeP achieving the best overall performance. In contrast, BLO predicts a static S-shape and fails to reproduce the multi-peaked structure. Furthermore, the Efficient Coding model predicts repulsion away from the modes (Appendix Figure 31), a pattern diametrically opposed to the attraction observed in human data. Overall, these results confirm **Prediction 2** and demonstrate that human probability perception involves a flexible prior distinct from resource allocation.

## 5 DISCUSSION AND CONCLUSION

Our results suggest that human probability distortion can be parsimoniously explained as optimal decoding from noisy neural encodings. Across tasks (dot counting, lottery pricing, lottery choice), we consistently find U-shaped encoding resources with peaks near 0 and 1. As we show in both theory and modeling, this nonuniformity induces systematic biases toward the interior of $[0, 1]$, providing a principled account of the classic S-shaped weighting function. Importantly, we find this account not only provides better fit than fixed probability weighting and the Bayesian Log-Odds Models, but also better fits than competing Bayesian accounts. Alternative explanations – such as regression away from response boundaries (Fennell & Baddeley, 2012; Bedi et al., 2025) or efficient coding with fully matched priors and encodings (Frydman & Jin, 2023) – fit the data less well. Our findings do not rule out efficient coding but suggest that priors and encodings may adapt on different timescales, preventing a full match of prior and encoding (Fritsche et al., 2020).

Interestingly, a similar S-shaped Bias and U-shaped Fisher Information also appear when probing vision–language models on the same task (Appendix C.16). Although the mechanisms in artificial systems are likely different, this observation provides an external point of comparison, and suggests that noisy encoding may reflect a more general information processing principle.

In conclusion, this work adds to a growing line of research linking decision-making distortions to imprecise yet structured mental representations (e.g. Woodford, 2012; Khaw et al., 2021; Frydman & Jin, 2022; Barretto-García et al., 2023; Zhang et al., 2020; Frydman & Jin, 2023). By grounding probability representation in Bayesian decoding of noisy encodings, our framework both unifies prior accounts and improves quantitative fit over the state-of-the-art Zhang et al. (2020) across both perceptual judgement and decision-making under risk.

## REPRODUCIBILITY STATEMENT

Complete derivations and proofs for the theoretical results are provided in Appendix B. All datasets used in Sections 4.1 and 4.2 are publicly available. Any preprocessing is described in Appendix C.2.

For modeling, we developed an implementation extending the optimization framework of Hahn & Wei (2024). Specific adaptations required for probability domain and task-specific processes are detailed in Appendix C.6, C.8, C.9.2. The reimplementation of Zhang et al. (2020) is detailed in Appendix C.11, C.12. Our scripts to reproduce all figures and experiments will be available upon publication.

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

APPENDIX

## A    THE USE OF LARGE LANGUAGE MODELS

We used a large language model (GPT-5) in two ways during the research and writing of this paper. First, it helped us with exploring the decision-making under risk choice dataset. Second, we used it to refine phrasing and to provide revision suggestions on draft manuscripts. Research ideas, modeling, and results analysis were conducted by the authors.

## B    THEORETICAL DERIVATIONS AND PROOFS

### B.1    THEORY ON FISHER INFORMATION (FI), BIAS AND MEAN SQUARE ERROR

#### B.1.1    PROPERTIES OF BAYESIAN MODEL

**Theorem 3** (Repeated from Theorem 1). *There are functions $A_{1,\sigma}, A_{2,\sigma}, A_{3,\sigma} : [0,1] \to \mathbb{R}$ that are bounded, positive, and satisfy*

$$A_{\dots,\sigma}(p) \leq \phi \left( \frac{\sigma}{\min\{|F(p) - F(0)|, |F(p) - F(1)|\}} \right) \tag{14}$$

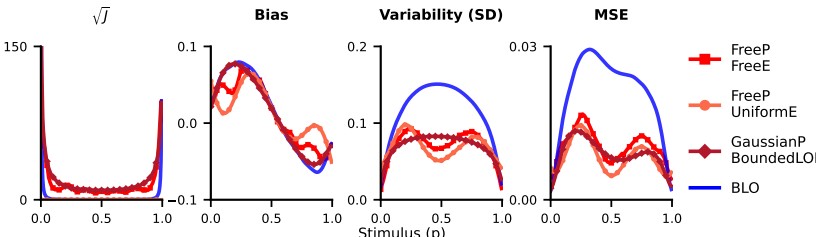

Figure 7: FI, Bias, Variance, MSE of the three models: Bayesian, BLO, Bayesian with uniform encoding, for parameters fitted in Section 4.1. BLO has an FI peaking at 0 and 1, as does the Bayesian model with log-odds encoding. All model variants predict a positive bias for small $p$, and a negative bias for large $p$, in line with the classical probability weighting function (Figure 1A). BLO and models with Gaussian priors predict a simple inverse U-shaped variability; other Bayesian priors can accommodate other variabilities.

*for some nondecreasing function $\phi : \mathbb{R} \to \mathbb{R}$ with $\lim_{x \downarrow 0} \phi(x) = 0$, across $p \in (0, 1); \sigma > 0$. Then, at any $p \in (0, 1)$, the Bayesian model has the bias, as $\sigma \to 0$:*

$$\text{Bias}(p) = \underbrace{A_{1,\sigma}(p) \cdot \frac{\text{sign}(0.5 - p)}{\sqrt{\mathcal{J}(p)}}}_{\text{Regression from Boundary}} + \underbrace{(1 - A_{2,\sigma}(p)) \cdot \frac{\text{d}}{\text{d}p}\left(\frac{1}{\mathcal{J}(p)}\right)}_{\text{Likelihood Repulsion}}$$

$$+ \underbrace{(1 - A_{3,\sigma}(p))\frac{1}{\mathcal{J}(p)} \cdot \frac{\text{d}}{\text{d}p}\left(\log P_{prior}(p)\right)}_{\text{Prior Attraction}} + \mathcal{O}(\sigma^4)$$

*Proof.* We first consider the case where $F(0) = -\infty$, $F(1) = \infty$, in which case (14) forces $A_{...,\sigma}(p) \equiv 0$. The above expression with $A_{...,\sigma}(p) \equiv 0$ simplifies to:

$$\underbrace{\frac{\text{d}}{\text{d}p}\left(\frac{1}{\mathcal{J}(p)}\right)}_{\text{Likelihood Repulsion}} + \underbrace{\frac{1}{\mathcal{J}(p)} \cdot \frac{\text{d}}{\text{d}p}\left(\log P_{prior}(p)\right)}_{\text{Prior Attraction}} + \mathcal{O}(\sigma^4)$$

Indeed, this is exactly the expression from Theorem 1 in Hahn & Wei (2024); the claim thus follows from that theorem.

We next consider the case where $F(0), F(1)$ are finite; we obtain this on the basis of Theorem 3 in Hahn & Wei (2024). That theorem examines the bias in the setting where the stimulus space has *one* boundary (e.g., $(-\infty, \theta_{Max}]$). For this case, it proves the decomposition (we consider the special case of $p = 2$, as we assume the posterior mean estimator):

$$\underbrace{-\frac{C_{1,p,D,F,\sigma}}{\sqrt{\mathcal{J}(p)}}}_{\text{Regression from Boundary}} + \underbrace{C_{3,p,D}\left(\frac{1}{\mathcal{J}(p)}\right)'}_{\text{Likelihood Repulsion}} + \underbrace{C_{2,p,D}\frac{1}{\mathcal{J}(p)}\left(\log P_{prior}(p)\right)'}_{\text{Prior Attraction}} + \mathcal{O}(\sigma^4) \quad (15)$$

where $D(p) := \frac{F(\theta_{Max}) - F(p)}{\sigma}$, and where $C_{...}$ are given as follows:

1. $C_{1,p,D,F,\sigma}$ depends on the probability $p$, the quantity $D(p)$, the encoding function $F$, and the sensory noise magnitude $\sigma$

2. $C_{2,p,D}$ depends on the probability $p$ and the quantity $D(p)$

3. $C_{3,p,D}$ depends on the probability $p$ and the quantity $D(p)$

4. $C_{1,p,D,F,\sigma}, C_{2,p,D}, C_{3,p,D}$ are positive everywhere

5. $C_{1,p,D,F,\sigma} = \Theta(1)$ when $\sigma \to 0$

6. $C_{1,\dots} \to_{D\to\infty} 0$, $C_{2/3,\dots} \to_{D\to\infty} 1$.

In fact, inspecting the proof of the theorem shows that $C_{3,p,D}, C_{2,p,D} \in (0,1)$ for all values of $p, D$, and that $C_{1,p,D(p),F,\sigma}$ is continuous in $p$ and $\sigma$ for $\sigma > 0$.

Our aim is to use this result to establish the bias in the setting where the stimulus space is $[0,1]$, i.e., has *two* boundaries.

Towards this end, we partition $(0,1)$ into $(0,1/2)$ and $(1/2,1)$.

First, $p \in (1/2, 1)$, the above decomposition is valid with $\theta_{Max} = 1$. The reason is because the contribution of the lower boundary at 0 to the bias can be absorbed into the $\mathcal{O}(\sigma^4)$ remainder, because only exponentially small probability mass of the likelihood of the encoding $m$ falls into regions outside a small local environment of $p$ (Hahn & Wei, 2024, SI Appendix, Section S.3.1.1).

Second, for $p \in (0, 1/2)$, by transforming the above result via $p \mapsto -p$, we obtain for the stimulus space $[\theta_{Min}, \infty)$ the bias:

$$\underbrace{\frac{C_{1,p,D,F,\sigma}}{\sqrt{\mathcal{J}(p)}}}_{\text{Regression from Boundary}} + \underbrace{C_{3,p,D}\left(\frac{1}{\mathcal{J}(p)}\right)'}_{\text{Likelihood Repulsion}} + \underbrace{C_{2,p,D}\frac{1}{\mathcal{J}(p)}\left(\log P_{prior}(p)\right)'}_{\text{Prior Attraction}} + \mathcal{O}(\sigma^4) \qquad (16)$$

where $D(p) := \frac{F(p) - F(\theta_{Min})}{\sigma}$, and where $C_{\dots}$ satisfy exactly the same conditions as given above.

Now write $\hat{C}_{\dots}$ for the coefficients for $p \in (1/2, 1)$ and $\tilde{C}_{\dots}$ for the coefficients for $p \in (0, 1/2)$; the same with $\hat{D}(p)$ and $\tilde{D}(p)$.

Now define

$$A_{1,\sigma}(p) := \hat{C}_{1,p,D(p),F,\sigma} \cdot 1_{p \geq 1/2} + \tilde{C}_{1,p,D(p),F,\sigma} \cdot 1_{p < 1/2}$$

$$A_{2,\sigma}(p) := 1 - \left(\hat{C}_{3,p,D(p)} \cdot 1_{p \geq 1/2} + \tilde{C}_{3,p,D(p)} \cdot 1_{p < 1/2}\right)$$

$$A_{3,\sigma}(p) := 1 - \left(\hat{C}_{2,p,D(p)} \cdot 1_{p \geq 1/2} + \tilde{C}_{2,p,D(p)} \cdot 1_{p < 1/2}\right)$$

First, inserting these definitions shows that the two decompositions do indeed produce the one claimed by the theorem.

We now need to show the remaining conclusions. By the assumptions above, $A_{1,\sigma}$ is positive and also continuous in both $\sigma$ and $p$. Next, by the facts that $C_{1,p,D,F,\sigma} = \Theta(1)$ when $\sigma \to 0$ and $C_{1,\dots} \to_{D\to\infty} 0$, we find that $A_{1,\sigma}(p)$ is bounded across $\sigma$ and $p$ whenever $\sigma$ is upper-bounded by some $\sigma_{Max}$. As $C_{2,\dots}, C_{3,\dots} \in [0,1]$, we find that $A_{2,\sigma}$ and $A_{3,\sigma}$ are positive and bounded. We have thus shown that $A_{1,\sigma}, A_{2,\sigma}, A_{3,\sigma}$ are positive and bounded.

Finally, as mentioned above, $\hat{C}_{1,p,\hat{D}(p),F,\sigma} \to 0$ as $\hat{D}(p) \to \infty$; equivalently,

$$\hat{C}_{1,p,\hat{D}(p),F,\sigma} = \mathcal{O}\left(\phi_1\left(\frac{1}{\hat{D}(p)}\right)\right) \qquad (17)$$

for some nondecreasing function $\phi_1$ with $\lim_{s\downarrow 0} \phi_1(s) = 0$. The same holds for $\tilde{C}, \tilde{D}$ with a function $\phi_2$. Thus,

$$A_{1,\sigma}(p) = \mathcal{O}\left(\phi_1\left(\frac{1}{\hat{D}(p)}\right)\right) + \mathcal{O}\left(\phi_2\left(\frac{1}{\tilde{D}(p)}\right)\right)$$

$$= \mathcal{O}\left(\max\left\{\phi_1(\frac{1}{\hat{D}(p)}), \phi_2(\frac{1}{\tilde{D}(p)})\right\}\right)$$

which is equivalent to the claim that

$$A_{1,\sigma}(p) \leq \phi\left(\frac{\sigma}{\min\{|F(p) - F(0)|, |F(p) - F(1)|\}}\right) \qquad (18)$$

across $p \in (0,1); \sigma > 0$, for some nondecreasing function $\phi$ with $\lim_{s \downarrow 0} \phi(s) = 0$. An analogous argument applies to $A_{2,\sigma}$ and $A_{3,\sigma}$.

$\square$

**Corollary 4.** *For unbounded log-odds encoding:*

$$\mathbb{E}[\hat{p}|p] - p = \underbrace{\sigma^2 p^2 (1-p)^2 \left(\log P_{prior}(p)\right)'}_{Attraction} + \underbrace{2\sigma^2 p(1-p)(1-2p)}_{Likelihood\ Repulsion} + \mathcal{O}(\sigma^4) \tag{19}$$

*Proof.* When $\beta = 0$, (11) simplifies to:

$$\sqrt{\mathcal{J}(p)} = \frac{1}{\sigma \cdot p \cdot (1-p)} \tag{20}$$

Furthermore, the coefficients $A_{...}$ vanish for this encoding, because the encoding map $F(p) = \int_{1/2}^{p} \sqrt{\mathcal{J}(q)} dq$ is the log-odds transformation, satisfying $F(0) = -\infty$, $F(1) = \infty$. Now

$$\frac{1}{\mathcal{J}(p)} = \sigma^2 \cdot p^2 \cdot (1-p)^2 \tag{21}$$

with derivative

$$\left(\frac{1}{\mathcal{J}(p)}\right)' = \sigma^2 \cdot 2 \cdot p \cdot (1-p) \cdot (1-2p) \tag{22}$$

Plugging these into Theorem 1 yields the result.

$\square$

**Theorem 5.** *At each $p \in (0,1)$, the Bayesian model has the response variability:*

$$\frac{1}{\mathcal{J}(p)} + \frac{2\sigma^2}{F'^2(p)} \frac{d}{dp} \underbrace{\left[\mathbb{E}[\hat{p}|p] - \mathbb{E}[F^{-1}(m)|p]\right]}_{Bias\ introduced\ by\ decoding} + \frac{\sigma^4 F''^2(p)}{2F'^6(p)} + O(\sigma^6) \tag{23}$$

*as $\sigma \to 0$.*

*Proof.* The term $\left[\mathbb{E}[\hat{p}|p] - \mathbb{E}[F^{-1}(m)|p]\right]$ corresponds to the quantity referred to as Decoding Bias in Hahn & Wei (2024); Hahn et al. (2025). The equation is then obtained from Lemma S24 in Hahn et al. (2025). Note that, for the quantity $C_{dec,M}$ used in that lemma, we have $[\mathbb{E}[\hat{p}|p] - \mathbb{E}[m|p]] = \sigma^2 C_{dec,M} + \mathcal{O}(\sigma^4)$, completing the proof. $\square$

**Theorem 6** (Optimality of Decoding). *The MSE of the decoded estimate in the Bayesian Model is*

$$\frac{1}{\mathcal{J}(p)} + O(\sigma^4) \tag{24}$$

*Proof.* Immediate, from $MSE = Var + Bias^2$, and noting that $Bias = \mathcal{O}(\sigma^2)$. $\square$

### B.1.2 PROPERTIES OF BLO MODEL

**Theorem 7** (Repeated from Theorem 2). *The BLO model in the limit of untruncated log-odds $(\Delta_+ \to \infty, \Delta_- \to -\infty)$, and a Bayesian model with unbounded log-odds encoding $(\beta = 0)$ and a specific unimodal prior $P_{prior}(p)$ (depending on $\Lambda_0$) have the same bias up to difference $\mathcal{O}(\kappa^2 + \sigma^4)$.*

*Proof.* We derive this as a corollary of Theorem 9. In the limit $\Delta_+ \to \infty, \Delta_- \to -\infty$, $\Phi(p)$ equals $p$. We now obtain the result by matching (19) and (29). Specifically, (29) then assumes the form:

$$\underbrace{\kappa \cdot p(1-p)V_p[\Lambda_0 - \Lambda(p)]}_{\equiv Attraction} + \underbrace{\frac{\sigma^2}{2} p(1-p)(1-2p)}_{\equiv Repulsion} + O(\kappa^2) + O(\sigma^4) \tag{25}$$

We match this with the Bayesian bias:

$$\underbrace{\sigma^2 p^2 (1-p)^2 \left(\log P_{prior}(p)\right)'}_{\text{Attraction}} + \underbrace{2\sigma^2 p(1-p)(1-2p)}_{\text{Likelihood Repulsion}} + \mathcal{O}(\sigma^4) \tag{26}$$

Under the identification $\kappa = \sigma^2_{Bayesian}$, $\sigma^2_{BLO} = 4\sigma^2_{Bayesian}$, we find

$$\left(\log P_{prior}(p)\right)' = \frac{V_p[\Lambda_0 - \Lambda(p)]}{p(1-p)} \propto \Lambda_0 - \Lambda(p) \tag{27}$$

Since $\Lambda(p)$ is monotonically increasing in $p$, this quantity is monotonically decreasing. Hence, $P_{prior}(p)$ has a single peak and describes a unimodal prior. $\qquad\square$

**Theorem 8.** *The BLO model has resource allocation $\sqrt{\mathcal{J}(p)}$ equal to:*

$$\frac{1}{\sigma} \cdot \left[ \left( -\kappa \frac{1-2p}{(1+\kappa p(1-p))^2} \right) \cdot (\Lambda(p) - \Lambda_0) + \omega_p \cdot \frac{\Psi}{(\Delta_+ - \Delta_-)/2} \begin{cases} 0 & if \ \lambda(p) \notin [\Delta_-, \Delta_+] \\ \lambda\prime(p) & else \end{cases} \right] \tag{28}$$

*Proof of Theorem 8.* Given the standard expression for the Fisher information of a Gaussian with parameter-dependent mean and constant variance, $\sqrt{\mathcal{J}(p)}$ equals

$$\frac{1}{\sigma} \cdot \frac{d\hat{\Lambda}_\omega(p)}{dp}$$

$$= \frac{1}{\sigma} \cdot \frac{d}{dp} \left[ \omega_p \cdot \Lambda(p) + (1-\omega_p) \cdot \Lambda_0 \right]$$

$$= \frac{1}{\sigma} \cdot \frac{d}{dp} \left[ \omega_p \cdot \Lambda(p) - \omega_p \cdot \Lambda_0 \right]$$

$$= \frac{1}{\sigma} \cdot \left[ \left( \frac{d}{dp} \omega_p \right) \cdot \Lambda(p) + \omega_p \left( \cdot \frac{d}{dp} \Lambda(p) \right) - \Lambda_0 \cdot \left( \frac{d}{dp} \omega_p \right) \right]$$

$$= \frac{1}{\sigma} \cdot \left[ \left( \frac{d}{dp} \omega_p \right) \cdot (\Lambda(p) - \Lambda_0) + \omega_p \left( \cdot \frac{d}{dp} \Lambda(p) \right) \right]$$

$$= \frac{1}{\sigma} \cdot \left[ \left( \frac{d}{dp} \frac{1}{1+\kappa p(1-p)} \right) \cdot (\Lambda(p) - \Lambda_0) + \omega_p \cdot \left( \frac{\Psi}{(\Delta_+ - \Delta_-)/2} \cdot \frac{d}{dp} \Gamma(\lambda(p)) \right) \right]$$

$$= \frac{1}{\sigma} \cdot \left[ \left( -\kappa \frac{1-2p}{(1+\kappa p(1-p))^2} \right) \cdot (\Lambda(p) - \Lambda_0) + \frac{\omega_p \cdot \Psi}{(\Delta_+ - \Delta_-)/2} \begin{cases} 0 & if \ \lambda(p) \notin [\Delta_-, \Delta_+] \\ \lambda\prime(p) & else \end{cases} \right]$$

$\qquad\square$

**Theorem 9.** *At any $p \in (0,1)$, the BLO model has the bias:*

$$\Phi(p) - p + \underbrace{\kappa \cdot \Phi(p)(1-\Phi(p))V_p[\Lambda_0 - \Lambda(p)]}_{\equiv Attraction} + \underbrace{\frac{\sigma^2}{2} \Phi(p)\left(1-\Phi(p)\right)\left(1-2\Phi(p)\right)}_{\equiv Repulsion} + O(\kappa^2) + O(\sigma^4) \tag{29}$$

*where $\Phi : (0,1) \to (0,1) : \Phi(p) := \lambda^{-1}(\hat{\Lambda}(p))$.*

*Proof of Theorem 9.* Consider the estimate as a function of $p$:

$$\hat{\pi}(p) = \lambda^{-1}(\hat{\Lambda}_\omega(p) + \epsilon_\lambda) \tag{30}$$

The bias is its expectation over $\epsilon_\lambda$ minus the true value:

$$\mathbb{E}\left[\lambda^{-1}(\hat{\Lambda}_\omega(p) + \epsilon_\lambda)\right] - p \tag{31}$$

where $\epsilon_\lambda \sim N(0, \sigma^2)$. To understand it, we perform a Taylor expansion around $\sigma = 0$, $\kappa = 0$. That is, we start by computing

$$\frac{\partial^2}{(\partial\epsilon_\lambda)^2}\left[\lambda^{-1}(\hat\Lambda_\omega(p) + \epsilon_\lambda)\right] = \lambda^{-1}(\Lambda(p))\left(1 - \lambda^{-1}(\Lambda(p))\right)\left(1 - 2\lambda^{-1}(\Lambda(p))\right) \tag{32}$$

at $\epsilon_\lambda = 0$, $\kappa = 0$, and

$$\frac{\partial}{\partial\kappa}\left[\lambda^{-1}(\hat\Lambda_\omega(p) + \epsilon_\lambda)\right] = \lambda^{-1}(\Lambda(p))\left(1 - \lambda^{-1}(\Lambda(p))\right)V_p\left[\Lambda_0 - \Lambda(p)\right] \tag{33}$$

at $\kappa = 0$, $\epsilon_\lambda = 0$.

Then

$$\begin{aligned}
\mathbb{E}\left[\lambda^{-1}(\hat\Lambda_\omega(p) + \epsilon_\lambda)\right] - p =& \kappa \cdot \frac{\partial}{\partial\kappa}\left[\lambda^{-1}(\hat\Lambda_\omega(p) + \epsilon_\lambda)\right] \\
& + \frac{\sigma^2}{2}\frac{\partial^2}{(\partial\epsilon_\lambda)^2}\left[\lambda^{-1}(\hat\Lambda_\omega(p) + \epsilon_\lambda)\right] \\
& + \lambda^{-1}(\Lambda(p)) \\
& - p \\
& + O(\kappa^2) + O(\sigma^4)
\end{aligned}$$

Filling in the above expressions, we get

$$\begin{aligned}
\mathbb{E}\left[\lambda^{-1}(\Lambda(p) + \epsilon_\lambda)\right] - p =& \kappa \cdot \lambda^{-1}(\Lambda(p))(1 - \lambda^{-1}(\Lambda(p)))V_p[\Lambda_0 - \Lambda(p)] \\
& + \frac{\sigma^2}{2}\lambda^{-1}(\Lambda(p))\left(1 - \lambda^{-1}(\Lambda(p))\right)\left(1 - 2\lambda^{-1}(\Lambda(p))\right) \\
& + \lambda^{-1}(\Lambda(p)) \\
& - p \\
& + O(\kappa^2) + O(\sigma^4)
\end{aligned}$$

$\square$

**Theorem 10.** *The BLO model has the response variability:*

$$\lambda^{-1}(\hat\Lambda_\omega(p))^2(1 - \lambda^{-1}(\hat\Lambda_\omega(p))^2\,\sigma^2 + \tfrac{1}{2}\lambda^{-1}(\hat\Lambda_\omega(p))^2(1 - \lambda^{-1}(\hat\Lambda_\omega(p)))^2(1 - 2\lambda^{-1}(\hat\Lambda_\omega(p)))^2\,\sigma^4$$
$$+ O(\sigma^6).$$

*Proof.* Consider the estimate as a function of $p$:

$$\hat\pi(p) = \lambda^{-1}(\hat\Lambda_\omega(p) + \epsilon_\lambda) \tag{34}$$

Conditioning on $p$, the variance over $\epsilon_\lambda$ is using a Taylor expansion:

$$\mathrm{Var}\left[\lambda^{-1}(\hat\Lambda_\omega(p) + \epsilon_\lambda)\right] = \left(f'(\mu)\right)^2\sigma^2 + \tfrac{1}{2}\left(f''(\mu)\right)^2\sigma^4 + O(\sigma^6).$$

where

$$f(x) := \lambda^{-1}(\hat\Lambda_\omega(p) + x) \tag{35}$$

Plugging this in, we obtain

$$\begin{aligned}
\mathrm{Var}\left[\lambda^{-1}(\hat\Lambda_\omega(p) + \epsilon_\lambda)\right] =& \lambda^{-1}(\hat\Lambda_\omega(p))^2(1 - \lambda^{-1}(\hat\Lambda_\omega(p))^2\,\sigma^2 \\
& + \tfrac{1}{2}\lambda^{-1}(\hat\Lambda_\omega(p))^2(1 - \lambda^{-1}(\hat\Lambda_\omega(p)))^2(1 - 2\lambda^{-1}(\hat\Lambda_\omega(p)))^2\,\sigma^4 \\
& + O(\sigma^6).
\end{aligned}$$

$\square$

## B.2 FI FOR EFFICIENT CODE IN FRYDMAN & JIN (2023)

Frydman & Jin (2023) take the encoding to be given by the code derived in Heng et al. (2020), which is defined via:

$$\theta(p) = \sin^2\left(\frac{\pi}{2}F(p)\right) \tag{36}$$

where $F(p)$ is the cumulative distribution function of the prior, whose density we denote $f(p)$. We now note, for $a(p) := \frac{\pi}{2}F(p)$:

$$\theta'(p) = 2\sin a \cos a \cdot a'(p) = \sin(2a) \cdot a'(p) \tag{37}$$

and hence $a'(p) = \frac{\pi}{2}f(p)$. Now for a code given by $m \sim \mathrm{Binomial}(n, \theta(p))$, the Fisher Information is

$$
\begin{aligned}
\mathcal{J}(p) =& \frac{n[\theta'(p)]^2}{\theta(p)(1-\theta(p))} \\
=& \frac{n\sin^2(2a)[a'(p)]^2}{\sin^2(a)(1-\sin^2(a))} \\
=& \frac{n\sin^2(2a)[a'(p)]^2}{\sin^2(a)\cos^2(a)} \\
=& \frac{n\sin^2(2a)[a'(p)]^2}{\frac{1}{4}\sin^2(2a)} \\
=& 4n[a'(p)]^2 \\
=& 4n(\frac{\pi}{2}f(p))^2 \\
\propto& f(p)^2
\end{aligned}
$$

We thus obtain $\sqrt{\mathcal{J}(p)} \propto p_{prior}(p)$.

## C EXPERIMENTAL METHODS AND SUPPLEMENTARY RESULTS

### C.1 THEORETICAL MAPPING OF MODEL VARIANTS

Table 1: Correspondence between existing theoretical accounts and the specific model variants used in our empirical comparison.

| Concept | Corresponding Model in Our Comparison |
|---|---|
| Fixed probability distortion function | LILO, Prelec |
| Log-odds based accounts | BLO |
| Regression based accounts | Bayesian model with uniform encoding |
| Efficient-coding | Bayesian model with prior-matched encoding |
| General Bayesian account | Bayesian model with freely fitted prior and encoding |

### C.2 DETAILS ON TASK AND DATASETS

#### C.2.1 JUDGMENT OF RELATIVE FREQUENCY (JRF) TASK IN SECTION 4.1

For this task, we analyzed the publicly available dataset from Zhang et al. (2020) and used Zhang & Maloney (2012) . On each trial, subjects were briefly shown an array of black and white dots and reported their estimate of the relative frequency of one color by clicking on a horizontal scale.

We used two datasets that differ in the granularity of the stimulus proportions:

- **Zhang et al. (2020) Dataset (JD, including JDA and JDB):**

– **subjects:** A total of 75 subjects were divided into two groups: 51 subjects who completed 660 trials (JDA) and 24 who completed 330 trials (JDB).

– **Stimuli:** Coarse-grained. The objective relative frequency was drawn from the 11 discrete probability levels: 0.01, 0.05, 0.1, 0.25, 0.4, 0.5, 0.6, 0.75, 0.9, 0.95, 0.99.

• **Zhang & Maloney (2012) Dataset (ZM12):**

– **subjects:** 11 subjects who completed 800 trials.

– **Stimuli:** Fine-grained. Stimulus proportions consisted of 99 levels, ranging from 0.01 to 0.99.

In both dataset, the total number of dots on any given trial was one of five values: 200, 300, 400, 500, or 600. For all datasets, each trial included the true stimulus proportion, the subject's estimate, total number of dots and other information, such as reaction time, that we didn't use for model fitting. We did not apply any data preprocessing.

### C.2.2 PRICING TASK IN DECISION-MAKING UNDER RISK (DMR) IN SECTION 4.2

For this task, we analyzed the publicly available dataset from Zhang et al. (2020). This task used the same procedure and design as Gonzalez & Wu (1999). On each trial, subjects were presented with a two-outcome monetary gamble (e.g., a 50% chance to win $100 or $0 otherwise) and a table of sure amounts. subjects made a series of choices between the gamble and the sure amounts, a process that used a sequential bisection method to narrow the range and determine their Certainty Equivalent (CE) for the gamble.

• **subjects:** The dataset comprises responses from the same 75 subjects who participated in the JD dataset in JRF task described above. 51 subjects from JDA performed 330 trials each and 24 subjects from JDB performed 165 trials each.

• **Stimuli:** The experiment used 15 distinct pairs of **non-negative** outcomes (e.g., $25 vs. $0; $100 vs. $50; $800 vs. $0). These were crossed with the same 11 probability levels from the JRF task for the higher outcome, resulting in 165 unique gambles.

• **Data:** Each recorded data point included the gamble's two outcomes, the probability of the higher outcome, the subject's final determined CE and other information that we didn't use for model fitting.

We did not apply any data preprocessing.

### C.2.3 CHOICE TASK IN DECISION-MAKING UNDER RISK (DMR) IN SECTION 4.2

For this task, we analyzed the publicly available dataset from the Choice Prediction Competition 2015 (CPC15). This study was designed to test and compare models on their ability to predict choices between gambles that elicit classic decision-making anomalies. On each trial, subjects were presented with two or more distinct monetary gambles and made a one-shot choice indicating their preference.

The full CPC15 dataset contains a wide range of gamble types, including ambiguous gambles (Amb=1), gambles with correlated outcomes (Corr $\neq$ 0), and multi-outcome lotteries (LotNum $>$ 1). For the purpose of our analysis, which requires two-outcome gambles with fully specified probabilities and independent payoffs, we applied the following filters:

• Amb = 0: excluded ambiguous gambles, i.e., those with probabilities not explicitly described to subjects.

• Corr = 0: excluded gambles with correlated payoffs across options.

• LotNum = 1: restricted to gambles with exactly two outcomes in each option.

This gives us a subset of the CPC15 dataset with following information:

• **subjects:** The subset contains responses from 153 subjects from the competition's estimation and test sets.

- **Stimuli:** The gambles covered 14 different behavioral phenomena, including the Allais and Ellsberg paradoxes. Unlike the pricing task, these gambles included both **gains and losses**, and the probabilities were drawn from a larger set of levels.

- **Data:** Each data point recorded the two gambles presented, the subject's choice, and information we didn't use for model fitting.

### C.3 DETAILS FOR ADAPTATION EXPERIMENT IN SECTION 4.3

This experiment used the JRF dot-counting paradigm, but critically, the distribution of stimulus proportions was manipulated to be bimodal to test the model's ability to adapt its prior. Data was collected on an online platform following the procedure in Zhang et al. (2020).

The bimodal dataset comprises responses from **26 subjects** across several designs to assess adaptation:

- 5 subjects performed 740 trials following from a bimodal distribution.

- 13 subjects performed an initial 740 trials following a bimodal distribution, followed by uniform trials to assess after-effects. In the 13 subjects, 7 subjects performed 840 trials(740 bimodal trials followed by 100 uniform trials), and 6 subjects performed 1136 trials(740 bimodal trials followed by 396 uniform trials).

- 8 subjects performed 942 bimodal trials interspersed with 198 uniform trials.

For all subjects in our collected datasets, in addition to the true number of black and white dots, the color designated for estimation, the subject's estimated proportion, and the reaction time, we also varies and logged the display time for each trial.

We removed estimated proportions of 0 and 100.

### C.4 FITTING MODELS TO DATASETS

Across all datasets, we ran all Bayesian models using the implementation from Hahn & Wei (2024); Hahn et al. (2025), using a grid of 200 points and regularization strength 1.0.

For all the Bayesian models, as well as the reimplemented Zhang et al. (2020)'s models, the optimization was performed using a gradient-based approach(Adam or SignSGD).

### C.5 DETAILS ON MODEL FIT METRICS

To compare the performance of all model variants, we use two primary evaluation metrics: summed Heldout $\Delta$NLL and summed $\Delta$AICc

#### C.5.1 SUMMED HELDOUT $\Delta$NLL

This metric measures a model's generalization performance, without penalizing for model complexity. The procedure of using this metric is as follows:

1. For each subject, the data is partitioned into a training set (9 out of 10 folds) and a held-out test set (the remaining 1 fold). A model is trained only on the training set.

2. The trained model is measured by calculating its Negative Log-Likelihood (NLL) on the held-out test set. A lower NLL indicates better predictions.

3. To compare models for that subject, we find the model with the lowest NLL (the best model). The $\Delta$NLL for any other model is its NLL minus the best model's NLL. The Summed Held-out $\Delta$NLL is the total of these individual $\Delta$NLL scores across all subjects.

#### C.5.2 SUMMED $\Delta$AICc

This is the metric used in Zhang et al. (2020). This metric assesses the overall quality of a model by balancing its performance with its simplicity. The procedure of using this metric is as follows:For

each subject, a model is trained on their entire dataset. This follows the procedure used in the work you are comparing against.

1. For each subject, a model is trained on all the trials. This follows the procedure used Zhang et al. (2020). We get the model's final NLL and count its number of free parameters ($k$).

2. We then calculate the Corrected Akaike Information Criterion (AICc) using the final NLL and $k$. The AICc formula essentially adds a penalty to the NLL for each extra parameter: AICc $\approx 2 * \text{NLL} + 2k$.

3. To compare models, we find the model with the lowest AICc for that subject. The $\Delta$AICc is a given model's AICc minus the best model's AICc. The Summed $\Delta$AICc is the total of these individual $\Delta$AICc scores across all subjects.

We report the Summed $\Delta$AICc metric primarily for parametric model variants, since it penalizes model complexity and is not straightforward to apply to nonparametric models.

For the nonparametric Bayesian models (e.g., FreeE, FreeP), defining the effective number of parameters is non-trivial due to regularization, which reduces the effective degrees of freedom below the raw grid size. To provide a lower bound on the model performance, we computed a maximally conservative AICc using the upper bound of model parameters. For example, we use 403 as the number of parameters of FreeE, FreeP model in JRF task(200 for encoding, 200 for prior, and 3 noise or mixing parameters). Because this assumption over-penalizes the nonparametric models, we do not include these values in the comparison bar plots. Instead, we report them in the corresponding footnotes.

## C.6 DETAILS ON CATEGORICAL AND ANALYTICAL FITTING

When modeling motor noise, there are two main ways to compute the likelihood of an observed value given the model-predicted $\hat{\theta}_m$. One way is using treat the response value as continuous, we refer to this way as analytical; the other way involves discretizing bins, which is referred to as categorical. In JRF and adaption data, we applied both versions in modelling motor noise in the proportion responses, and in DMR pricing task, we applied the same idea to the Certainty Equivalent (CE) responses.

### C.6.1 JRF AND ADAPTATION TASK

**Analytical Version.** We treat responses as continuous and assume they are drawn from a Gaussian centered on the Bayesian estimate $\hat{p}(m)$ with variance $\sigma_{\text{motor}}^2$. Because responses are bounded by the grid $[r_{\min}, r_{\max}]$, the Gaussian is truncated and normalized using the corresponding CDF values. This gives the exact continuous likelihood, though it can be numerically unstable when the motor variance is very small. Finally, we mix this motor likelihood with a uniform component to account for guessing:

$$Z = \Phi\left(\frac{p_{\max} - \hat{p}(m)}{\sigma_{\text{motor}}}\right) - \Phi\left(\frac{p_{\min} - \hat{p}(m)}{\sigma_{\text{motor}}}\right),$$

$$P(p_{\text{obs}} \mid m) = \frac{1}{Z} \frac{1}{\sqrt{2\pi\sigma_{\text{motor}}^2}} \exp\left(-\frac{(p_{\text{obs}} - \hat{p}(m))^2}{2\sigma_{\text{motor}}^2}\right),$$

$$P_{\text{mix}}(p_{\text{obs}} \mid m) = (1-u)P(p_{\text{obs}} \mid m) + u.$$

**Categorical Version.** Here we discretize the response axis into bins $\{c_j\}_{j=1}^{J}$, compute a categorical distribution over bins for each $\hat{p}(m)$, and assign each observed response to its nearest bin. With a fine grid and the bin-width correction, this converges to the analytical solution, but remains stable at very small motor variance:

$$\log P(p_j \mid m) = \text{logsoftmax}\left(-\frac{(p_j - \hat{p}(m))^2}{2\sigma_{\text{motor}}^2}\right),$$

$$j^{(p_{\text{obs}})} = \arg\min_j |p_{\text{obs}} - p_j|,$$

$$\log \tilde{P}(p_{\text{obs}} \mid m) = \log P(p_j \mid m) - \log \Delta c,$$

$$P_{\text{mix}}(p_{\text{obs}} \mid m) = (1-u)\tilde{P}(p_{\text{obs}} \mid m) + u \cdot \tfrac{1}{J}.$$

In this case, the number of bins is the same the number of the grid size we discretize the input stimuli. We used 200 for the JRF and Adapation datasets.

### C.6.2 DMR Pricing Task

**Analytical Version.** We treat the CE report as a continuous variable and assume it is drawn from a Gaussian centered on the model prediction $\mu_m$ with variance $\sigma_{\text{motor}}^2$. The likelihood of an observed CE is given directly by this Gaussian density. This provides the exact continuous likelihood, though it can become numerically unstable when $\sigma_{\text{motor}}^2$ is very small. As before, we mix this motor likelihood with a continuous uniform distribution to account for guessing:

$$P(\text{CE}_{\text{obs}} \mid m) = \exp\left(-\tfrac{1}{2}\left[\frac{(\text{CE}_{\text{obs}} - \mu_m)^2}{\sigma_{\text{motor}}^2} + \log\left(2\pi\sigma_{\text{motor}}^2\right)\right]\right)$$

$$P_{\text{mix}}(\text{CE}_{\text{obs}} \mid m) = (1-u) \cdot P(\text{CE}_{\text{obs}} \mid m) + u \cdot \frac{1}{\text{CE}_{\text{max}} - \text{CE}_{\text{min}}}.$$

**Categorical Version.** Here we discretize the response axis into bins $\{c_j\}_{j=1}^J$, model a categorical distribution over bins for each m, and then select the probability of the observed bin. With a fine grid and the -$\log \Delta c$ correction, the categorical method converges to the analytical one:

$$\log P(\text{CE}_j \mid m) = \text{logsoftmax}\left(-\frac{(\hat{p}_m - \text{CE}_j)^2}{2\sigma_{\text{motor}}^2}\right)$$

$$j^{(\text{CE}_{\text{obs}})} = \arg\min_j |\text{CE}_{\text{obs}} - \text{CE}_j|$$

$$\log \tilde{P}(\text{CE}_{\text{obs}} \mid m) = \log P(\text{CE}_j \mid m) - \log \Delta c$$

$$P_{\text{mix}}(\text{CE}_{\text{obs}} \mid m) = (1-u)\tilde{P}(\text{CE}_{\text{obs}} \mid m) + u \cdot \tfrac{1}{J}.$$

In Zhang et al. (2020)'s dataset, $\text{CE}_{\text{max}} = 800$ and $\text{CE}_{\text{min}} = 0$. We use 1000 grid size for the categorical version.

### C.7 Details on Bayesian Model for JRF Task

### C.7.1 Bayesian Model Variants

We tested several variants of our Bayesian framework by combining different priors and encodings; some are discussed in Section 4.1:

- **Priors:**
  - **Uniform Prior (UniformP):** This variant assumes a uniform distribution of prior over the range of possible stimuli(i.e., across all grid points). There is no learnable parameter.
  - **Gaussian Prior (GaussianP):** This variant assumes a Gaussian distribution over the range of possible grid points. The two fitted parameters are Gaussian mean and Gaussian standard deviation.
  - **Freely Fitted Prior (FreeP):** In this variant, all values of the prior distribution across 200 grid points are treated as trainable parameters. While this allows the model maximal flexibility to fit the data, the large number of trainable parameters can make the calculation of summed $\Delta$AICc challenging. There are 200 freely fitted parameters.

**Priors**

- **Encodings**

    - **Uniform Encoding (UniformE):** This encoding assumes a uniform distribution of encoding over the range of stimuli. There is no fitted parameter.

    - **Fixed Unbounded Log-Odds Encoding (UnboundedLOE):** The encoding is proportional to the log-odds of the stimuli(grid values). This is consistent with the log-odds assumption underlying Zhang's unbounded log-odds models. There is no fitted parameter.

    - **Bounded Log-Odds Encoding (BoundedLOE):** This encoding is given by $F'(p) = \frac{1}{(p+\beta_1)(1-p+\beta_2)}$[5], where $x$ is the value of grid, and $\beta_1$ and $\beta_2$ are small, positive, learnable parameters between 0 and 1, which we bound using the sigmoid function. This form is motivated by Zhang's BLO models and our discussion in Section 3.2. There are two fitted parameters: $\beta_1$ and $\beta_2$.

    - **Prior-matched Encoding(PriorMatchedE):** This encoding assumes that the resources is proportional to the prior distribution, i.e., the encoding density is identical to the prior. There are no additional fitted parameters beyond those of the prior.

    - **Freely Fitted Encoding (FreeE):** Similar to freely fitted prior. There are 200 freely fitted parameters.

Parameters for each Bayesian model variant, including prior parameters, encoding parameters, sensory noise variance, motor variance and mixture logit are optimized against the subject-level data using the same gradient based method described for BLO models on the same task.

### C.7.2 Performance Comparison of All Model variants

In this part, we show the performance of model variants on both evaluation metrics(Summed Heldout $\Delta$NLL and $\Delta$AICc) and with both analytical and categorical fitting methods in Figure8 and 9[6]. The Bayesian model variants with red color in the figures are detailed in Appendix Section C.7.1. The model variants from Zhang et al. (2020) with blue color in the figures are detailed in Appendix Section C.11.2.

Overall, our model performance better than Zhang et al. (2020)'s model variants across two metrics and two fitting approaches. Within Bayesian models, models using a bounded log-odds encoding outperform those with an unbounded encoding, and a Gaussian prior is superior to a uniform prior. Within Zhang et al. (2020)'s model variants, bounded model performs better than unbounded model, and using $V(p)$(explained in Eq 41) is better than assuming a constant value $V$.

---

[5]Note that this is equivalent to (11) in the setting where different $\beta$'s, written here as $\beta_1, \beta_2$, are allowed for positive and negative counts, up to an (irrelevant, as it doesn't depend on $p$) proportionality constant:

$$\frac{1}{\sigma}\left(\frac{1}{p+\beta_1} + \frac{1}{(1-p)+\beta_2}\right) \propto \frac{1+\beta_2+\beta_1}{(p+\beta_1)(1-p+\beta_2)} \propto \frac{1}{(p+\beta_1)(1-p+\beta_2)} \qquad (38)$$

[6]Note that we report the Summed $\Delta$AICc metric only for parametric model variants, since it penalizes model complexity and is not straightforward to apply to nonparametric models.

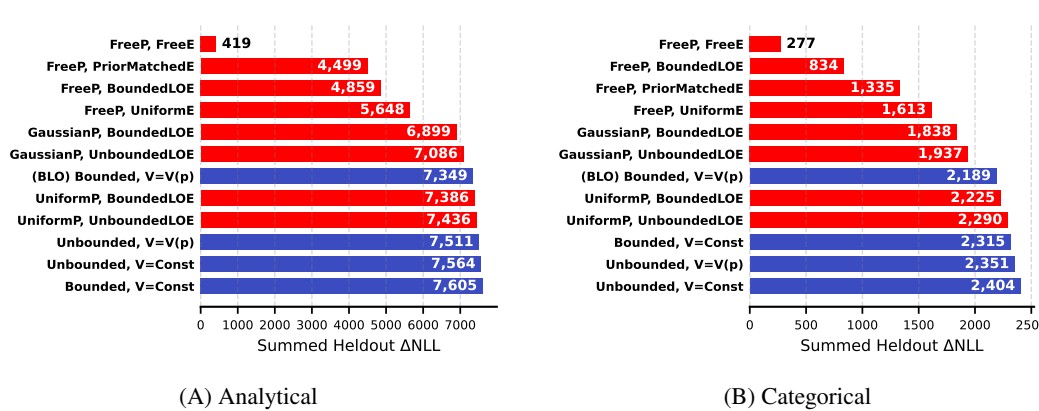

(A) Analytical

(B) Categorical

Figure 8: **JRF Task:** Performance of the models, measured by the Summed Heldout ΔNLL metric. See Appendix C.7.1 for the shorthands for Bayesian models (red). We refer to Zhang et al. (2020) for the shorthands for their model variants (blue). Analytical and categorical fitting methods are explained in Appendix C.6. The results shown in the main text correspond to Analytical. This result is mentioned in main text section 4.1

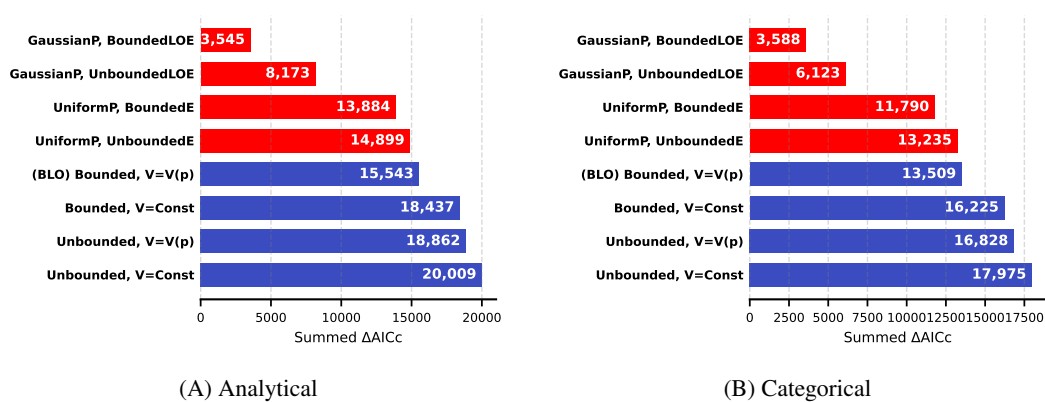

(A) Analytical

(B) Categorical

Figure 9: **JRF Task:** Performance of the models, measured by the Summed ΔAICc metric. See Appendix C.7.1 for the shorthands for Bayesian models (red). We refer to Zhang et al. (2020) for the shorthands for their model variants (blue). Analytical and categorical fitting methods are explained in Appendix C.6. The results shown in the main text correspond to Analytical. The summed ΔAICc values in this plot are shown for parametric variants. We also computed this metric for the nonparametric FreeP, FreeE model using a maximally conservative penalty ($k = 403$, see Appendix C.5). This model outperformed the best parametric model in both fitting methods: **For (A) Analytical:** The FreeP, FreeE model's total Summed AICc (87,118.56) is the lowest among all the model variants and is substantially lower than the best-performing parametric model, GaussianP, BoundedLOE (183,880.57). **For (B) Categorical:** Similarly, the FreeP, FreeE model's total Summed AICc (90,091.29) is the lowest and is lower than the best-performing parametric model, GaussianP, BoundedLOE (97,190.31).

### C.7.3 ANALYSIS OF FITTED PRIOR AND RESOURCES

Figure 10 plots the fitted resources for our Bayesian model (red) and the BLO model (blue) for 86 subjects in the JRF task. For nearly all subjects, the resources from both models are U-shaped, with peaks near the probability endpoints of $p = 0$ and $p = 1$.

A closer examination reveals a difference. As shown in Figure 11, which restricts the y-axis for clarity, the resources of the BLO model exhibit two points of discontinuity. This finding is formally predicted by and consistent with our Theorem 8.

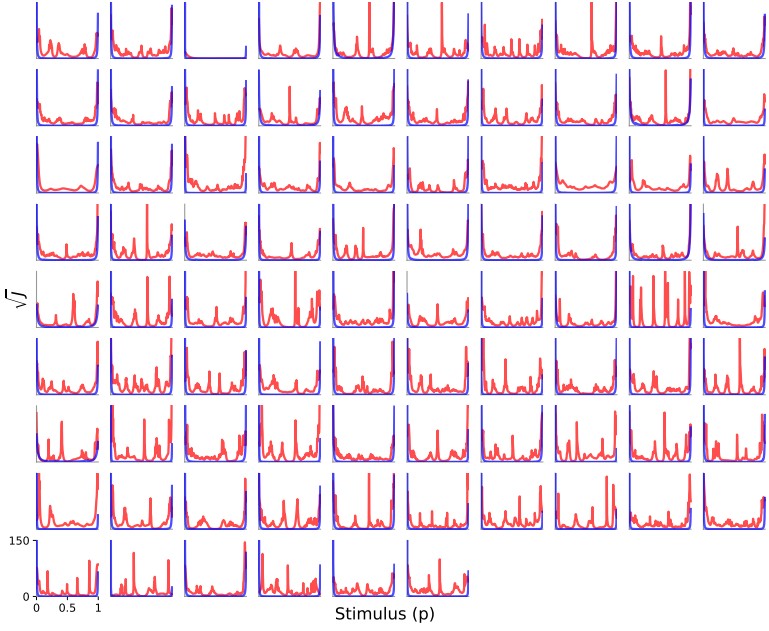

Figure 10: **JRF Task:** Per-subject resources ($\sqrt{J(p)}$) for the Bayesian model with freely fitted prior(in red) and encoding and the BLO model(in blue). Subjects S1 to S51 are from dataset JDA, S52 to S75 from dataset JDB (both from Zhang et al. (2020)), and S76 to S86 from dataset ZM12 (Zhang & Maloney (2012)).

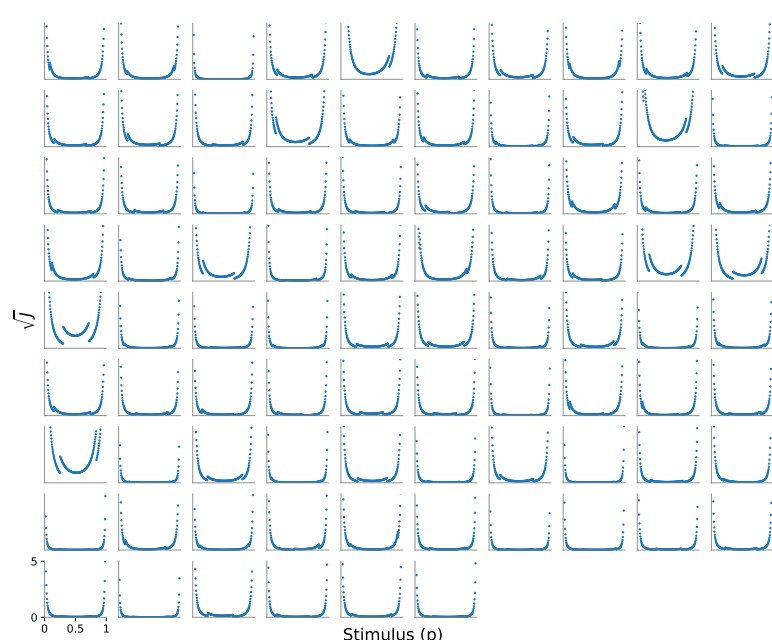

Figure 11: **JRF Task:** Discontinuity observed in the resources ($\sqrt{J(p)}$) of the BLO model. The proof of this property is in Theorem 8. Subjects S1 to S51 are from dataset JDA, S52 to S75 from dataset JDB (both from Zhang et al. (2020)), and S76 to S86 from dataset ZM12 (Zhang & Maloney (2012)).

Figure 12 compares the group-level fitted priors for the Bayesian model variants. The parametric Gaussian prior is shown to capture the main features of the non-parametric, freely fitted prior. The freely fitted prior with matched encoding shows not only peaks at 0.5, but also at 0 and 1, which is a property of the Resources.

Figure 13 compares the group-level resources for all Bayesian model variants. The resources are U-shaped for the parametric models (the Bayesian log-odds variants), and the freely fitted resources also recover this U-shape.

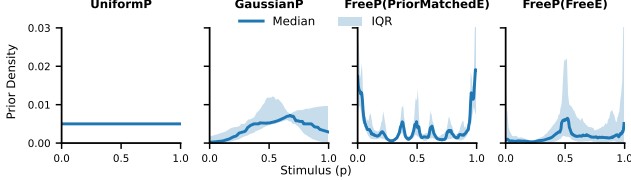

Figure 12: **JRF Task:** Group-level priors for the four prior components evaluated in this task. The solid line shows the group median, and the shaded area indicates the interquartile range (IQR). Plotting details are provided in Appendix C.14.

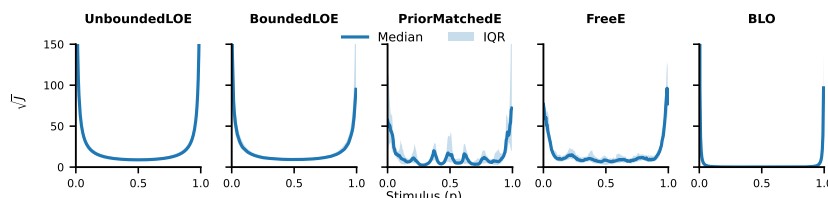

Figure 13: **JRF Task:** Group-level resources ($\sqrt{J(p)}$) for the four encoding components evaluated in this task, along with the resources of the BLO model. The solid line shows the group median, and the shaded area indicates the interquartile range (IQR). Plotting details are provided in Appendix C.14. This result is mentioned in main text section 4.1.

### C.7.4 ANALYSIS OF BIAS AND VARIANCE

Figure 33 decompose the bias of four Bayesian model variants into attraction and repulsion. Both attractive and repulsive components point away from 0 and 1. Both are needed to account for the overall distortion, as both prior and encoding need to be nonuniform to achieve good model fit.

Figures 14 and 15 show the per-subject bias and variance, respectively, providing a more detailed view of the group-level results presented in the main text (Figure 3). The methods used to calculate these quantities are detailed in Appendix C.15.

The per-subject plots confirm the main findings. For bias, both the Bayesian model and the BLO model capture the bis pattern of the non-parametric estimates. The key divergence appears in the response variability. Figure 15 clearly shows that while our Bayesian model captures subject-level variability, the BLO model consistently fails to model the dip at $p = 0.5$.

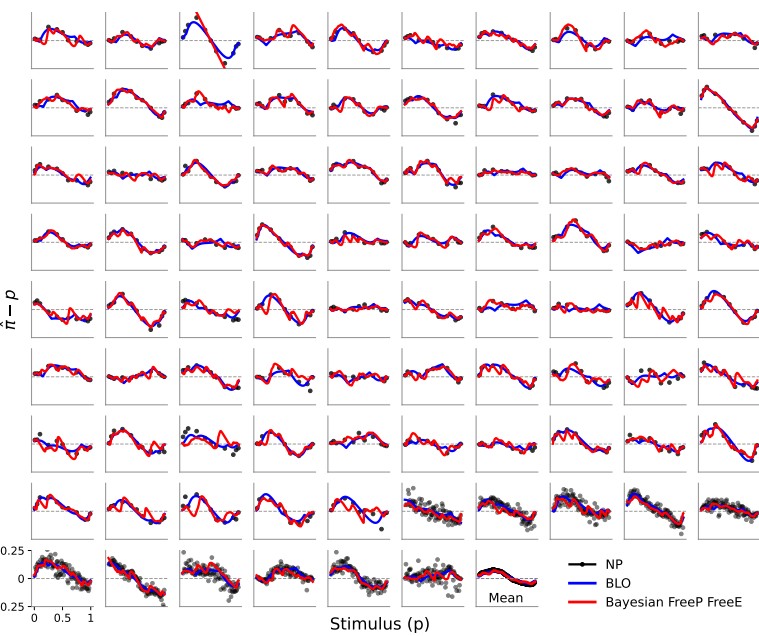

Figure 14: **JRF Task:** Per-subject bias of the non-parametric estimates from the data, the BLO model and the Bayesian model with free prior and free encoding. Subjects S1 to S51 are from dataset JDA, S52 to S75 from dataset JDB (both from Zhang et al. (2020)), and S76 to S86 from dataset ZM12 (Zhang & Maloney (2012)).

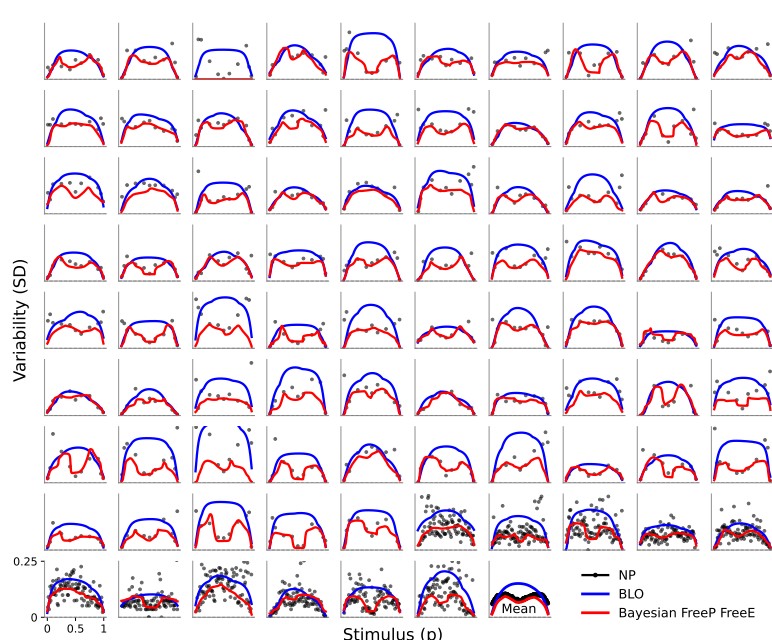

Figure 15: **JRF Task:** Per-subject variability of the non-parametric estimates from the data, the BLO model and the Bayesian model with free prior and free encoding. Subjects S1 to S51 are from dataset JDA, S52 to S75 from dataset JDB (both from Zhang et al. (2020)), and S76 to S86 from dataset ZM12 (Zhang & Maloney (2012)).

## C.8   DETAILS ON BAYESIAN MODEL FOR DMR PRICING TASK

### C.8.1   BAYESIAN MODEL VARIANTS AND FITTING PROCDURE

We tested the similar Bayesian model variants for the pricing task as we did for the JRF task (detailed in Appendix C.7.1). However, fitting these models to the Certainty Equivalent (CE) data required a specific two-stage procedure. The goal was to first convert each observed CE into an "implied" subjective probability weight, and then fit our Bayesian models to these weights.

**Stage 1: Deriving trial-by-trial implied estimates.**   The goal of this initial stage was to convert each raw CE response into a non-parametric, trial-level estimate of the subjective probability weight, $\hat{\pi}_{\mathrm{implied},t}(p)$. For each trial $t$, we fits a free variable. We also fit $\alpha$ for applying the CPT utility function. To account for additional variability in CE, we included an extra noise term $\epsilon_{\mathrm{CE}}$ and optimized parameters by minimizing the loss between predicted and observed CE. This method is similar to that of Zhang et al. (2020), but our implementation works on a trial-by-trial basis rather than on 11 discrete probability levels.

**Stage 2: Fitting Bayesian estimator.**   The set of $(p_t, \hat{\pi}_{\mathrm{implied},t})$ pairs derived from Stage 1 was then used as the target data to fit the parameters of our Bayesian model. These parameters in turn determine the set of optimal point estimates, $\hat{p}(m)$ (the decoded stimulus value for each possible internal measurement m), by maximizing the likelihood of the implied weights.

**Stage 3: Final Likelihood Maximization on Original CE Data.**   To ensure a fair and direct comparison with the BLO model, the final model evaluation was based on the likelihood of the original CE data. In this final stage, the Bayesian encoding parameters (and therefore the set of $\theta(m)$ values) were held fixed from the results of Stage 2. We then performed a final optimization to find the subject's remaining parameters—the utility exponent $\alpha$ and the CE noise variance $\sigma_{\mathrm{CE}}$—that maximized the log-likelihood of their observed CE responses. The resulting maximum log-likelihood value was then used to calculate the Held-out $\Delta$NLL and $\Delta$AICc scores.

**Model variants.** Because stage 2 closely resembles the JRF task, we applied the same set of model variants used there (Appendix C.7.1), with the exception of the Fixed Unbounded Log-Odds Encoding. We excluded this variant to focus on encoding schemes that showed better performance in this task.

### C.8.2 PERFORMANCE COMPARISON OF ALL MODEL VARIANTS

Figure 16 and Figure 17 presents the performance of model variants on both evaluation metrics.

For the likelihood of the observed CE data, we chose to present results from a categorical likelihood function in the main text. While a fully analytical (continuous Gaussian) likelihood is possible—and in our tests, this analytical version of our model achieves a lower summed $\Delta$heldout loss than Zhang's models—we opted for the categorical approach. We argue that the DMR task, which involves a comparative judgment, is inherently more categorical in nature than the continuous estimation required in the JRF task. To ensure a fair and direct comparison, we re-evaluated Zhang's original models using this identical categorical likelihood.

Across both evaluation methods, the Bayesian models (red bars) consistently outperformed Zhang et al.'s variants (blue bars). In particular, when measured by summed $\Delta$NLL, the Bayesian models achieved substantially smaller losses, indicating a much better account of the observed CE distributions. The advantage is also evident under the summed $\Delta$AICc, where Bayesian models again dominate.

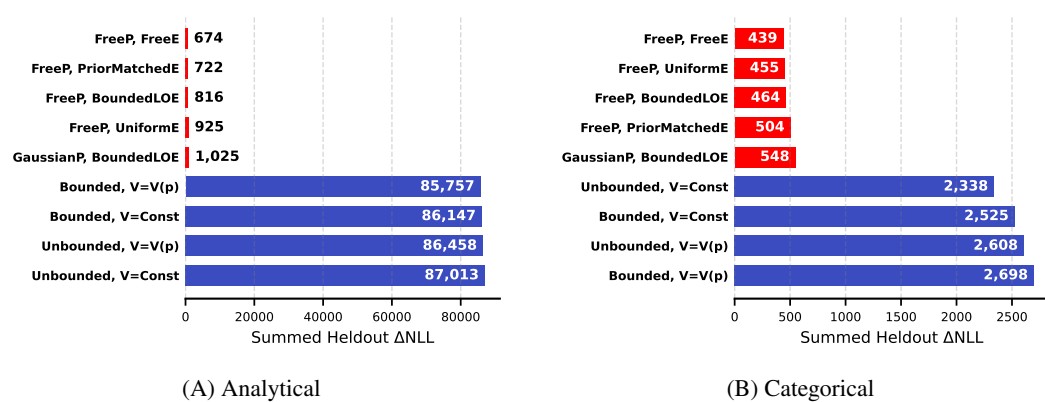

(A) Analytical

(B) Categorical

Figure 16: **DMR Pricing Task:** Performance of the models, measured by the Heldout $\Delta$NLL metric. See Appendix C.7.1 for the shorthands for Bayesian models (red). We refer to Zhang et al. (2020) for the shorthands for their model variants (blue). Analytical and categorical fitting methods are explained in Appendix C.6. The results shown in the main text correspond to Categorical.

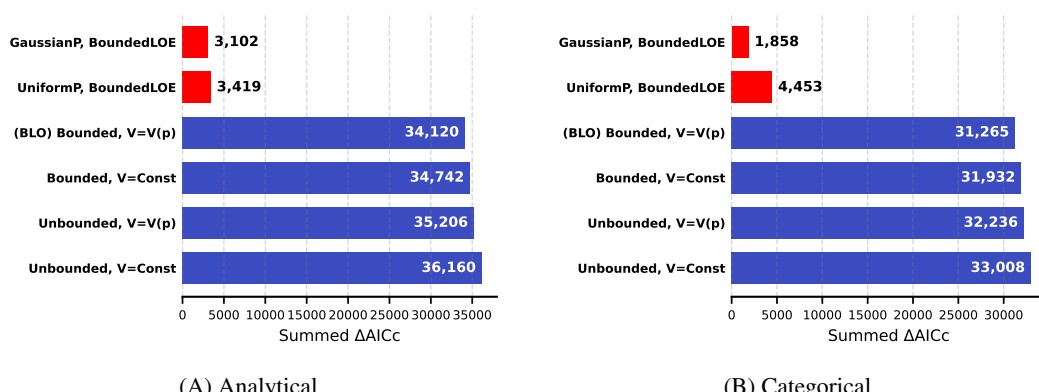

(A) Analytical          (B) Categorical

Figure 17: **DMR Pricing Task:** Performance of the models, measured by the $\Delta$AICc metric. See Appendix C.7.1 for the shorthands for Bayesian models (red). We refer to Zhang et al. (2020) for the shorthands for their model variants (blue). Analytical and categorical fitting methods are explained in Appendix C.6. The results shown in the main text correspond to Categorical. The summed $\Delta$AICc values in this plot are shown for parametric variants. We also computed this metric for the nonparametric FreeP, FreeE model using a maximally conservative penalty ($k = 406$, see Appendix C.5). This model outperformed the best parametric model in both fitting methods: **For (A) Analytical:** The FreeP, FreeE model's total Summed AICc (0) is the lowest among all the model variants and is substantially lower than the best-performing parametric model, GaussianP, Bounded-LOE (205,089.75). **For (B) Categorical:** Similarly, the FreeP, FreeE model's total Summed AICc (0) is the lowest and is substantially lower than the best-performing parametric model, GaussianP, BoundedLOE (196,520.55).

### C.8.3 ANALYSIS OF FITTED PRIOR AND RESOURCES

The fitted resources in Figure 18 closely resemble those obtained in the JRF task. The BLO model doesn't have meaningful resources because it doesn't fit a noise in the encoding phase.

For the prior, the freely fitted version exhibits a shape similar to the Gaussian prior, which accounts for the strong performance of Bayesian models with Gaussian priors reported in Section C.8.2.

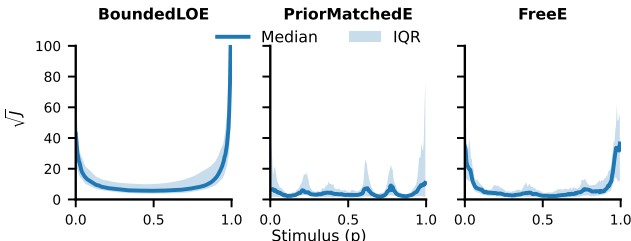

Figure 18: **DMR Pricing Task:** Group-level resources ($\sqrt{J(p)}$) for the three encoding components evaluated in this task. The solid line shows the group median, and the shaded area indicates the interquartile range (IQR). Plotting details are provided in Appendix C.14.

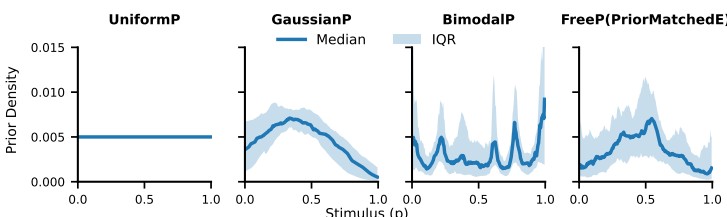

Figure 19: **DMR Pricing Task:** Group-level priors for the four prior components evaluated in this task. The solid line shows the group median, and the shaded area indicates the interquartile range (IQR). Plotting details are provided in Appendix C.14. Free fitting is compatible with a unimodal prior.

### C.8.4 Analysis of Bias and Variance

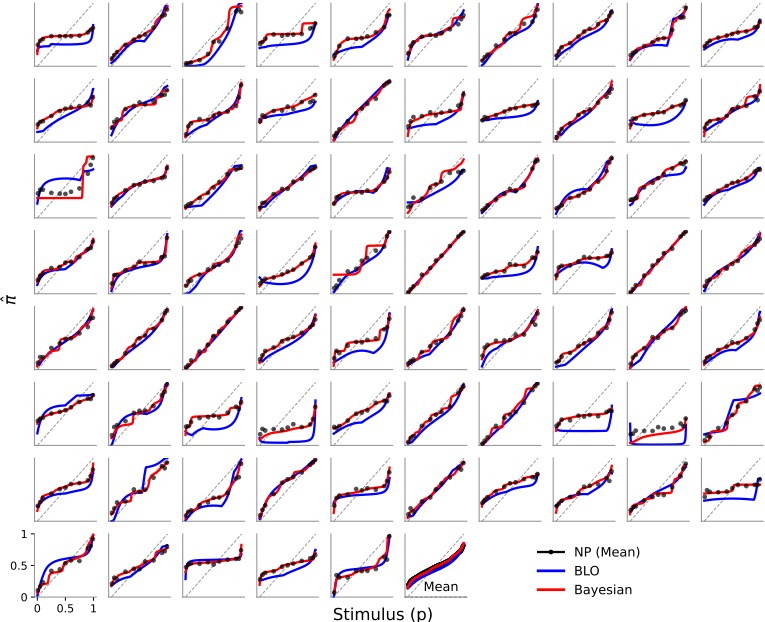

Figure 20: **DMR Pricing Task:** Per-subject bias of the non-parametric estimates from the data, the BLO model and the Bayesian model with free prior and free encoding on the DMR pricing task. Subjects S1 to S51 are from dataset JDA, and S52 to S75 are from dataset JDB (both from Zhang et al. (2020)).

We present the per-subject bias of the probability estimate, $\hat{\pi}$, for both the Bayesian and BLO models. As the figures show, both models capture the general pattern of bias for most subjects. A direct comparison of the variance of $\hat{\pi}$ is not shown because the two models treat this quantity fundamentally differently. The BLO model's estimate $\hat{\pi}_{\text{BLO}}$ is a deterministic point value and thus has zero variance. In contrast, our Bayesian estimate, $\hat{\pi}_{\text{Bayesian}}$, is the mean of a full posterior distribution and has inherent sensory noise variance. The methods used to calculate the non-parametric, BLO's, and Bayesian model's bias are detailed in Appendix C.15.

### C.9 DETAILS ON BAYESIAN MODEL FOR DMR CHOICE TASK

#### C.9.1 FULL UTILITY FUNCTION AND LOGIT CHOICE RULE

**Full Utility Function**

$$\text{Utility} = \begin{cases} w^+(p)v(X) + (1 - w^+(p))v(Y) & \text{if } X \geq Y \geq 0 \\ (1 - w^-(1 - p))v(X) + w^-(1 - p)v(Y) & \text{if } X > Y \text{ and } X, Y < 0 \\ w^+(p)v(X) + w^-(1 - p)v(Y) & \text{if } X > 0 > Y \end{cases} \tag{39}$$

This equation is mentioned in main text section 4.2

**Logit Choice Rule**

$$P(\text{Choose B}) = \text{sigmoid}(\tau \cdot (\text{Utility}_B - \text{Utility}_A)) \tag{40}$$

The parameter $\tau$ ($\tau > 0$,) controls the choice sensitivity and is a free parameter to fit.

#### C.9.2 BAYESIAN MODEL VARIANTS AND FITTING PROCEDURE

We tested several model variants:

- **Priors:**
  - **Freely Fitted Prior:** As we mainly focus on validating the resources shape with this task, we only evaluate the freely fitted prior, which is the same as described in Appendix C.7.1.
- **Encodings:**
  - **Bounded Log-odds Encoding, Prior-matched Encoding, Freely Fitted Encoding:** Same as in Appendix C.7.1.

Because the dataset includes both gains and losses, we fit separate probability weighting functions for them. Specifically, we estimate separate priors for gains and for losses, while assuming a shared encoding across both. The resulting probability weighting functions are then computed as the expectation of $\hat{p}$ under the encoding distribution.

#### C.9.3 PERFORMANCE COMPARISON OF ALL MODEL VARIANTS

Figure 21 shows the performance of different models on the DMR choice task. Among all tested models, the Bayesian variant with freely fitted prior and encoding (FreeP, FreeE) achieve the best fit, clearly outperforming both other Bayesian variants and classical parametric weighting functions. Models with parametric encoding perform worse, and parametric models such as LILO and Prelec also show substantially higher $\Delta$AICc. The Bayesian model with uniform encodoing performs the worst, showing that this encoding cannot capture the probability distortion in this task.

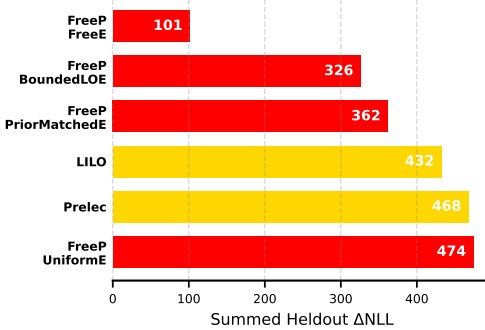

Figure 21: **DMR Choice Task:** Performance of the models, measured by the Heldout $\Delta$NLL metric. See Appendix C.7.1 for the shorthands for Bayesian models (red).

We also computed the summed $\Delta$AICc metric for the nonparametric models using a maximally conservative penalty ($k = 403$, see Appendix C.5). The freely fitted model's total Summed AICc(335,287.37) is the lowest among all the model variants and is substantially lower than the best-performing parametric model Prelec(484,724.40).

### C.9.4 Analysis of Fitted Prior and Resources

We show the group-level encoding resources (Figure 22) and priors (Figure 23). Across models, the fitted resources consistently U-shaped, with peaks at 0 and 1. Interestingly, the freely fitted priors for both gains and losses are also U-shaped.

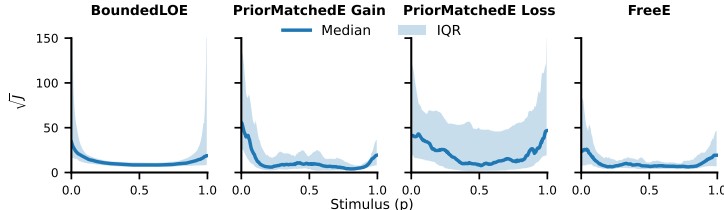

Figure 22: **DMR Choice Task:** Group-level resources ($\sqrt{J(p)}$) for the four encoding components evaluated in this task. The solid line shows the group median, and the shaded area indicates the interquartile range (IQR). Plotting details are provided in Appendix C.14.

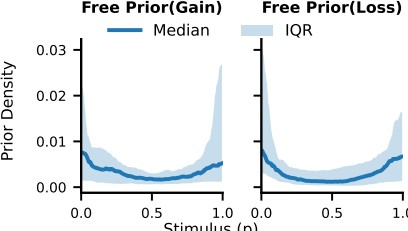

Figure 23: **DMR Choice Task:** Group-level prior of the Bayesian model with freely fitted priors and encoding. The solid line shows the group median, and the shaded area indicates the interquartile range (IQR). Plotting details are provided in Appendix C.14.

### C.9.5 Analysis of Bias

Figure 24 shows the per-subject bias for the Bayesian freely fitted model, the LILO model, and the Prelec model. Across subjects, the three models tend to capture similar patterns of bias, although the detailed shapes at the individual level likely reflect a considerable degree of overfitting.

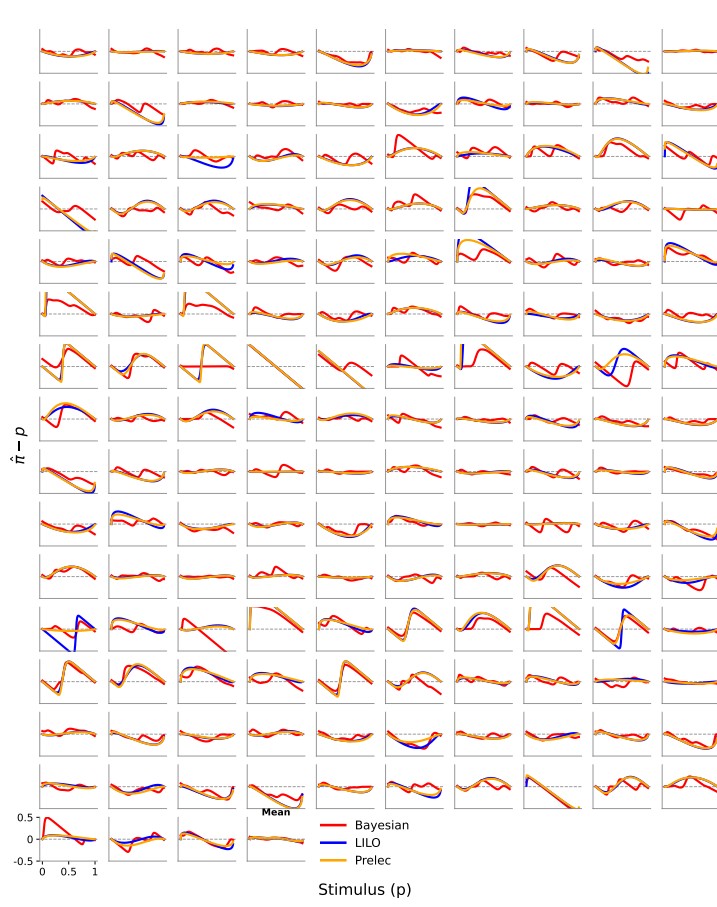

Figure 24: **DMR Choice Task:** Per-subject bias of the LILO model, the Prelec model and the Bayesian model with free prior and free encoding on the DMR pricing task. Subjects S1 to 153 are from dataset CPC15 (Erev et al., 2017).

### C.10 DETAILS ON BAYESIAN MODEL FOR ADAPTATION BIMODAL DATA

#### C.10.1 BAYESIAN MODEL VARIANTS

We tested several model variants. Most aspects were the same as the variants used for the JRF task, with the exception that here we evaluated a bimodal prior instead of a uniform prior.

- **Priors:**
    - **Bimodal Prior:** This variant assumes a bimodal distribution over the range of possible grid points. The free parameters are the two means and their corresponding standard deviations.
    - **Gaussian Prior, Freely Fitted Prior:** These variants are the same as described in Appendix C.7.1.
- **Encodings:**
    - **Unbounded Log-odds Encoding, Bounded Log-odds Encoding, Prior-matched Encoding, Freely Fitted Encoding:** Same as in Appendix C.7.1.

#### C.10.2 PERFORMANCE COMPARISON OF ALL MODEL VARIANTS

For the Adaptation Task, model comparison again shows a clear advantage for Bayesian variants over BLO. When measured by both held-out $\Delta$NLL (Figure 25) and $\Delta$AICc (Figure 26), Bayesian

models consistently achieve substantially lower scores, indicating a better quantitative account of the observed responses.

Among the Bayesian models, those with bimodal priors provide a closer fit than their Gaussian prior counterparts, reflecting the bimodal structure of the stimulus distribution. The freely fitted prior yields the best performance overall, suggesting that allowing the prior to flexibly adapt to the empirical distribution of stimuli gives the most accurate description of subjects' behavior. We also find that model with matched prior and encoding perform particularly well.

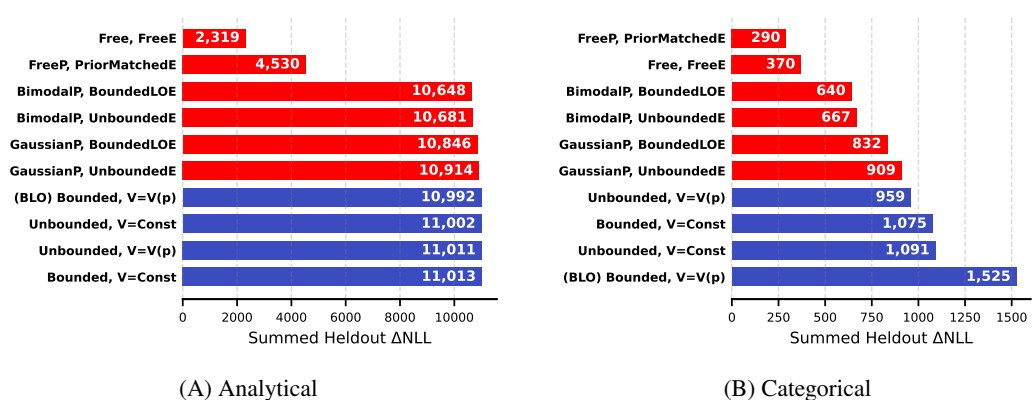

(A) Analytical

(B) Categorical

Figure 25: **Adaptation Task:** Performance of the models, measured by the Heldout $\Delta$NLL metric. Stimuli follows bimodal distribution. See Appendix C.7.1 for the shorthands for Bayesian models (red). We refer to Zhang et al. (2020) for the shorthands for their model variants (blue). Analytical and categorical fitting methods are explained in Appendix C.6. The results shown in the main text correspond to Analytical.

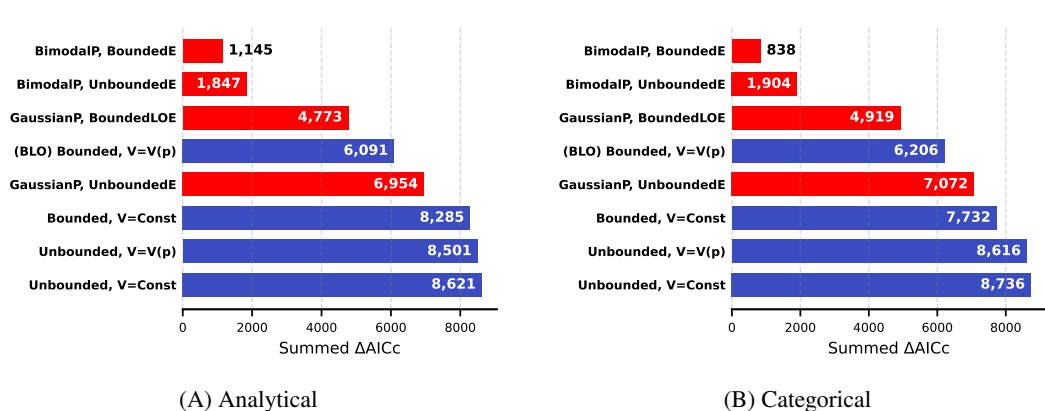

(A) Analytical

(B) Categorical

Figure 26: **Adaptation Task:** Performance of the models, measured by the Heldout $\Delta$AICc metric. Stimuli follows bimodal distribution. See Appendix C.7.1 for the shorthands for Bayesian models (red). We refer to Zhang et al. (2020) for the shorthands for their model variants (blue). Analytical and categorical fitting methods are explained in Appendix C.6. The results shown in the main text correspond to Analytical. The summed $\Delta$AICc values in this plot are shown for parametric variants. We also computed this metric for the nonparametric FreeP, FreeE model using a maximally conservative penalty ($k = 403$, see Appendix C.5). This model outperformed the best parametric model in both fitting methods: **For (A) Analytical:** The FreeP, FreeE model's total Summed AICc (13,274.42) is the lowest among all the model variants and is substantially lower than the best-performing parametric model, bimodalP, BoundedLOE (181,887.62).

### C.10.3 ROBUSTNESS OF MODEL COMPARISON VIA BOOTSTRAPPING

We performed a subject-level bootstrap analysis on the aggregate model comparison results (Figure 6D). Specifically, we resampled the 26 subjects with replacement 10,000 times. For each bootstrap sample, we calculated the Summed Held-out $\Delta$NLL difference between our freely fitted Bayesian model (FreeP, FreeE) and each competing model.

Table 2 summarizes the results. We report the Bootstrap Support, the percentage of resampled datasets in which the FreeP, FreeE model achieved a lower Summed NLL than the competitor. We also report the 95% Confidence Interval (CI) of the difference in NLL (competing model minus FreeP, FreeE).

The freely fitted Bayesian model significantly outperformed all competing accounts, including the Efficient Coding model (PriorMatchedE) and the BLO model, achieving a bootstrap support greater than 98%. The 95% confidence intervals for the NLL differences exclude zero for all comparisons. This analysis confirms the superiority of our Bayesian account in this task.

Table 2: **Adaptation Task:** Bootstrap analysis of model comparison in the (10,000 iterations). The freely fitted Bayesian model (FreeP, FreeE) is the reference. Positive $\Delta$NLL values mean the reference model fits better. CI is Confidence Interval.

| Competitor Model Variants | Bootstrap Support | 95% CI of Difference ($\Delta$NLL) |
|---|---|---|
| FreeP, PriorMatchedE (Efficient Coding) | 98.1% | $[233.43, 7550.74]$ |
| BimodalP, BoundedLOE | 100.0% | $[7177.88, 13032.97]$ |
| BimodalP, UnboundedE | 100.0% | $[7216.18, 13054.51]$ |
| GaussianP, BoundedLOE | 100.0% | $[7380.49, 13224.09]$ |
| (BLO) Bounded, $V = V(p)$ | 100.0% | $[7518.55, 13399.81]$ |
| GaussianP, UnboundedE | 100.0% | $[7466.34, 13271.41]$ |
| Bounded, $V = \text{Const}$ | 100.0% | $[7532.97, 13426.86]$ |
| Unbounded, $V = V(p)$ | 100.0% | $[7518.60, 13437.80]$ |
| Unbounded, $V = \text{Const}$ | 100.0% | $[7510.38, 13429.20]$ |

### C.10.4 ANALYSIS OF FITTED PRIOR AND RESOURCES

Figure 27 shows that the group-level resources largely retain the characteristic U-shape across encoding variants, consistent with the JRF task results. This indicates that encoding efficiency remains highest near the extremes of the probability scale. Because each subject was exposed to a different bimodal stimulus distribution, we do not plot a group-level prior. Instead, Figure 28 shows the freely fitted prior for each subject, which for most subjects successfully adapts to the underlying bimodal distribution.

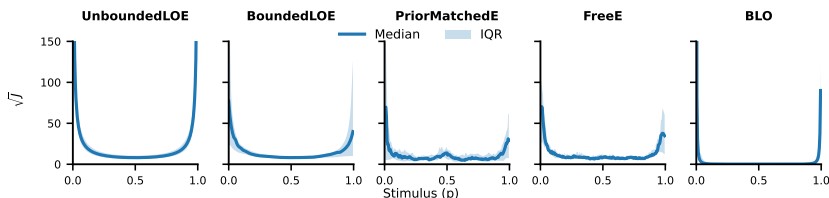

Figure 27: **Adaptation Task:** Group-level resources ($\sqrt{J(p)}$) for the three encoding components evaluated in this task. The solid line shows the group median, and the shaded area indicates the interquartile range (IQR). Plotting details are provided in Appendix C.14.

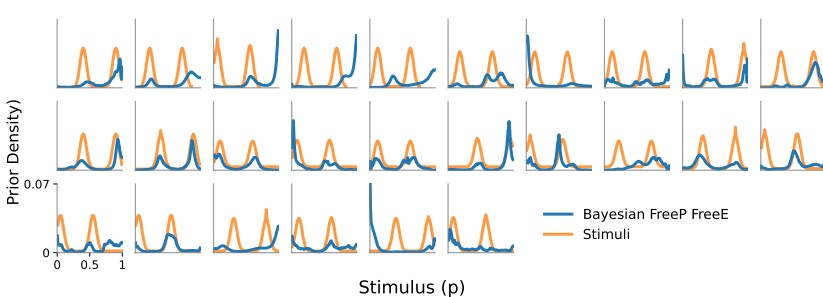

Figure 28: **Adaptation Task:** Fitted priors for individual subjects. The freely fitted prior (blue) successfully adapts to the bimodal stimulus distribution (orange).

### C.10.5 ANALYSIS OF BIAS AND VARIANCE

Figure 29 shows the bias of each subject. The BLO model captures broad trends in bias but lacks the flexibility to account for the more complex, stimulus-dependent bias patterns observed in human data. In contrast, the Bayesian model with freely fitted prior and encoding provides a closer fit to subject-level biases, particularly in regions where deviations from linearity are more pronounced.

In the variability figure(Figure 30)The Bayesian model also provides a superior account of variability compared to BLO. While BLO could capture the overall variability magnitude, the Bayesian model models both the overall magnitude and the shape of variability more accurately, and aligns more closely with the human data.

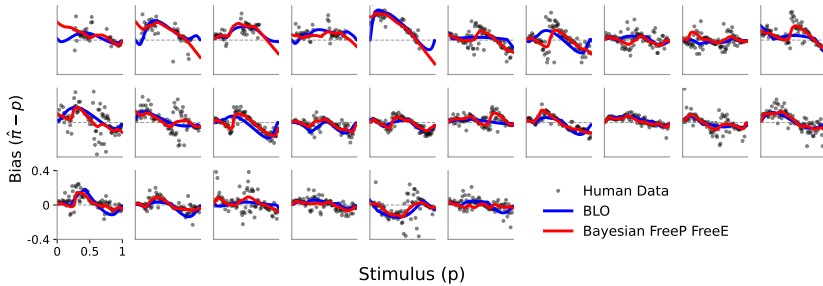

Figure 29: **Adaptation Task:** Per-subject bias of the non-parametric estimates from the data, the BLO model and the Bayesian model with free prior and free encoding on the Adaptation task. Subjects S1 to S26 are from our collected dataset.

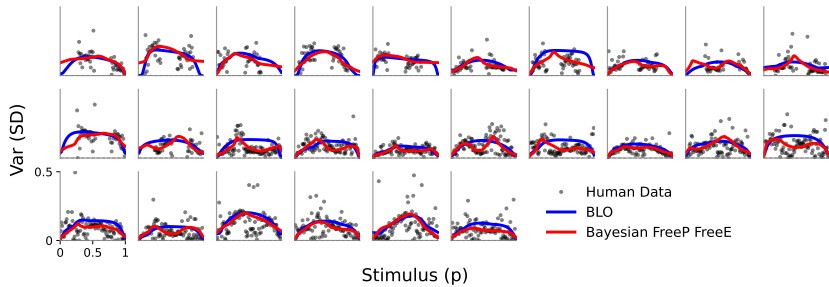

Figure 30: **Adaptation Task:** Per-subject variability of the non-parametric estimates from the data, the BLO model and the Bayesian model with free prior and free encoding on the Adaptation task. Subjects S1 to S26 are from our collected dataset.

Figure 31 illustrates how different models account for biases under bimodal stimulus statistics. Human subjects show systematic attraction toward the two stimulus modes: bias is positive on the left side of each peak and negative on the right, producing a multi-peaked structure. The Bayesian model with freely fitted prior and encoding closely matches this pattern, indicating that flexible prior–encoding combinations can capture the adaptation to bimodal input. By contrast, when the encoding is constrained to match the prior, the model predicts biases pointing away from the prior modes, a hallmark of efficient coding, which diverges from the empirical data. Finally, the BLO model yields a relatively monotone bias that fails to reproduce the bias of the human responses. Together, these results show that BLO and efficient-coding-based predictions are insufficient to explain behavior, whereas the general Bayesian framework with flexible priors and encodings provides a close fit.

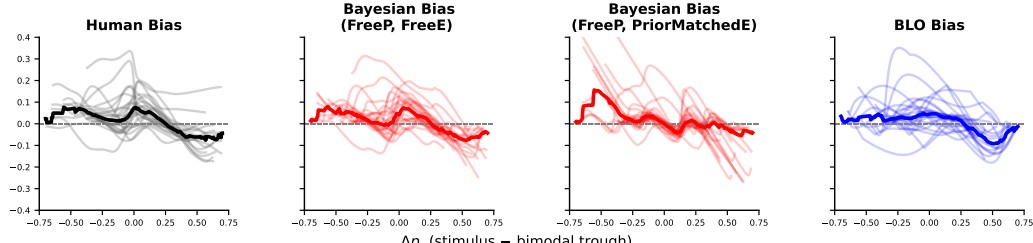

Figure 31: **Adaptation Task:** Bias under bimodal stimulus statistics. Each panel shows estimation bias as a function of $\Delta p$ (stimulus proportion relative to the trough between the two stimulus peaks). Thin curves show individual subjects or model fits; thick curves show the across-subject or across-fit median. (Left) Human data: subjects exhibit multi-peaked biases, with attraction toward both stimulus modes. (Second) Bayesian model with freely estimated prior and encoding (FreeP, FreeE) successfully reproduces this multi-peaked structure. (Third) Bayesian model with FreeP but encoding constrained to match the prior (PriorMatchedE) predicts biases away from the prior modes, consistent with efficient coding but inconsistent with human data. (Right) BLO model fails to capture the human pattern. This figure is dicussed in the main text section 4.3

## C.11 Details on BLO Model for JRF Task and Adapatation Data

### C.11.1 Application of BLO Model to JRF Task

While initially developed for decision under risk (DMR) tasks, the BLO model can be adapted to account for perception in judgment of relative frequency (JRF) tasks. In the JRF context, the core principles of bounded log-odds encoding, truncation, and variance compensation remain, but the model additionally accounts for sensory noise introduced by the observer's sampling of stimuli.

In JRF tasks, it is assumed that people do not process every element in the display. Instead, they sample a subset of items from the whole display. This sampling introduces additional noise in the observer's estimate. Consequently, the variance of the sample-based estimate, $V(\hat{p})$, is modeled with a finite population correction: If the total number of items is $N$ and the observer samples $n_s$ items, the variance of the sample-based estimate is given by:

$$V(\hat{p}) = \frac{p(1-p)}{n_s} \cdot \frac{N - n_s}{N - 1} \tag{41}$$

, where $N$ is the total number of dots and $n_s$ is a free parameter. We note that sampling of the items is not implemented in the published implementation from Zhang et al. (2020), and we correspondingly do not attempt to model it in our reimplementation of the BLO model.

### C.11.2 Reimplementation of BLO and LLO Models Variants

We reimplemented the key model variants proposed by Zhang et al. (2020) for JRF. Specifically, we focused on four variants, Bounded Log-Odds(BLO) and Linear Log-Odds (LLO) models, each combined with either a constant perceptual variance (V=Const) or a proportion-dependent variance compensation (V=V(p)).

Following Zhang et al. (2020), we fit these models to the data per subject by minimizing the negative log-likelihood of the observed responses given the model's predicitions. Our reimplementation achieved nearly identical estimated parameter values and model fits to those reported by Zhang et al. (2020) for JRF.

For all these reimplemented models, the optimization was performed using a gradient-based approach(Adam or SignSGD) rather than the original Nelder-Mead optimizer(fminsearchbnd) used in their implmentation. We set parameter bounds for our optimization based on Zhang et al. (2020)'s settings. the hyperbolic tangent (tanh) function was applied to parameters such as $\Delta^-$ and $\Delta^+$ , while the exponential function was used for strictly positive parameters like $\kappa$. Parameters not naturally constrained were optimized directly.

We successfully achieved nearly identical estimated parameter values and model fits to those reported by Zhang et al. (2020) for the dot counting task, thereby validating our reimplementation.

### C.12  DETAILS FOR BLO ON DMR PRICING TASK

#### C.12.1  APPLICATION OF BLO MODEL TO DMR TASK

The uncertainty or variance associated with this internal encoding is modeled as being proportional to the binomial variance.

$$V(\hat{p}) \propto p(1-p) \tag{42}$$

Following Zhang et al. (2020), the estimate $\hat{\pi}(p)$ is integrated into Cumulative Prospect Theory (CITE) to predict choice behavior; the certainty equivalent (CE) for a two-outcome lottery $(x_1, p; x_2, 1-p)$ is given by:

$$\text{CE} = U^{-1}\left[U(x_1) \cdot \hat{\pi}(p) + U(x_2) \cdot (1 - \hat{\pi}(p))\right] + \varepsilon_{CE} \tag{43}$$

Here, $U(\cdot) = x^\alpha$ is the utility function, and $\varepsilon_{CE}$ represents Gaussian noise on the CE scale.

#### C.12.2  REIMPLEMENTATION OF BLO AND LLO MODELS VARIANTS

We first reimplemented the four main parametric models proposed by Zhang et al. (2020) for the DMR task. These included the BLO and LLO models, each combined with either a proportion-dependent variance compensation (V(p)) or a constant variance (Const V). As with the JRF task analysis, these models were fitted to the data for each subject individually. Our implementation utilized gradient-based optimizers (specifically, Adam or SignSGD) for parameter optimization.

We observed that the negative log-likelihood (NLL) values obtained from our reimplemented models were nearly identical to Zhang's reported fitted results for most variants. However, for the BLO + V(p) model, our reimplemented model showed, on average, a 7.45 higher loss (negative log-likelihood) compared to Zhang's original results; though this does not impact quantitative model comparison with the Bayesian model.

### C.13  CROSS-TASK ANALYSIS OF RESOURCES AND BIAS

#### C.13.1  FREELY FITTED PRIOR AND RESOURCES IN JRF AND DMR TASKS

The freely fitted priors and corresponding resources functions in three tasks are shown in Figure 32). Priors differ noticeably between tasks, but resources consistently exhibits the U-shape, with highest sensitivity near the extremes. Interestingly, in the DMR Choice tasks, the shape of the prior appears more closely aligned with the resources than in the other tasks. This prior–resources match is reminiscent of the account proposed by Frydman & Jin (2023), but the fact that it is not observed across all tasks suggests that such alignment is limited in generality.

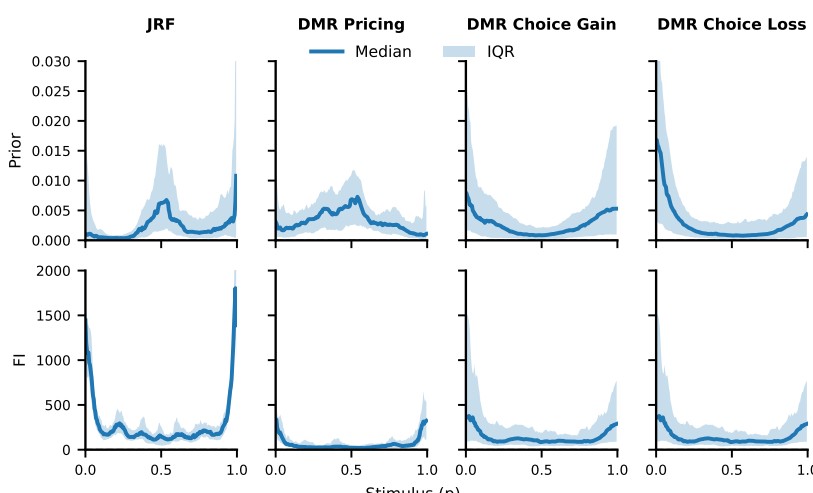

Figure 32: **Across Tasks:** Freely fitted priors (top) and resources $\sqrt{J}$ (bottom) across tasks. The solid line shows the group median, and the shaded area indicates the interquartile range (IQR). Plotting details are provided in Appendix C.14. This figure is mentioned in main text section 4.1.

### C.13.2 BIAS DECOMPOSITION IN ALL TASKS

As shown in Section 3.1, we decomposed the total estimation bias ($\mathbb{E}[\hat{p}|p] - p$) into three components: Prior Attraction, Likelihood Repulsion, and Boundary Regression. We computed these components by:

**Regression from Boundary** We computed this as the difference in total bias between the original constrained model and a model with extended boundaries. In the model with extended boundaries, we extended the stimulus space from $[0, 1]$ to a larger interval $[-1, 2]$ to remove boundary effects. The encoding function $F(p)$ was extended by continuing the resources constantly from its values at the endpoints $p = 0$ and $p = 1$.

**Likelihood Repulsion** We computed this component as the bias arising under the same model with the extended stimulus space and the prior replaced by a uniform prior. In this model, both prior attraction and boundary regression are removed, isolating the repulsion effect.

**Prior Attraction** We computed it as the difference between the mode of the posterior distribution and the expected value of the sensory likelihood function.

In Figure 33, the decomposition reveals two critical insights regarding the origin of probability distortion. First, regarding the failure of Uniform Encoding, the most distinct qualitative difference appears in the Likelihood Repulsion column. The model with uniform encoding exhibits zero likelihood repulsion. Consequently, to approximate the empirical S-shaped bias, it is forced to rely exclusively on Boundary Regression and a heavily distorted Prior Attraction, making it structurally insufficient to generate the smooth, global S-shaped distortion observed in human data. Second, the trade-off between bias and variability fit. The Gaussian prior produces a smooth attraction curve. While this fits the general shape of the bias well, the rigidity of the Gaussian form prevents it from forming the sharp peak necessary to capture the dip in response variability at $p = 0.5$. The nonparametric priors exhibit fluctuations in their attraction. This distortion arises because of a sharp prior peak to capture the low variability at $p = 0.5$ 1. This sharp peak exerts excessive local attraction, creating slight deviations in the bias curve to satisfy the global normalization constraint.

Figure 34 illustrates the bias decomposition for the DMR Pricing task. Similar to JRF, the UniformE model exhibits zero likelihood repulsion. It attempts to compensate for this absence by inducing a heavily distorted Prior Attraction and relying on Boundary Regression. While this compensation allows for a moderate quantitative fit in pricing (a task where we compare the expected value instead of probability), it remains mechanistically distinct from the repulsion in U-shaped models, ann we seek for more evidence in Figure 35, the bias decomposition for the DMR Choice task. As in other tasks, the Uniform model shows zero likelihood repulsion. However, unlike in the Pricing task, the Prior Attraction here remains relatively flat and fails to compensate for the missing repulsion. This flatness likely occurs because, in a binary choice task, increasing prior precision (to reduce variance) without sufficient sensory discriminability at the boundaries would only compress values and reduce the model's ability to predict preference reversals. The model thus defaults to a smoother prior to avoid introducing excessive bias error.

Figure 36 shows how the Bayesian model captures the shift in behavior under bimodal stimulus statistics. In Prior Attraction, the GaussianP model produces a simple, smooth attraction curve. In contrast, both the FreeP and BimodalP models exhibit multi-peaked attraction patterns. These fluctuations correspond to attraction forces directed toward the two modes of the bimodal stimulus distribution.

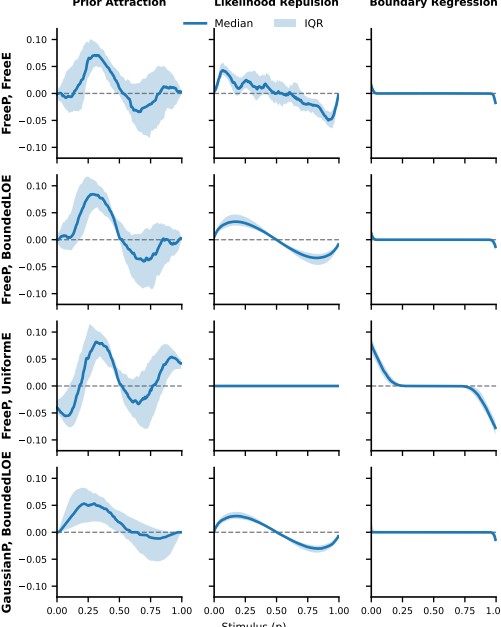

Figure 33: **JRF Task:** The comparison between bias decomposition of four Bayesian model variants in Figure 3. The solid line shows the group median, and the shaded area indicates the interquartile range (IQR). Plotting details are provided in Appendix C.14.

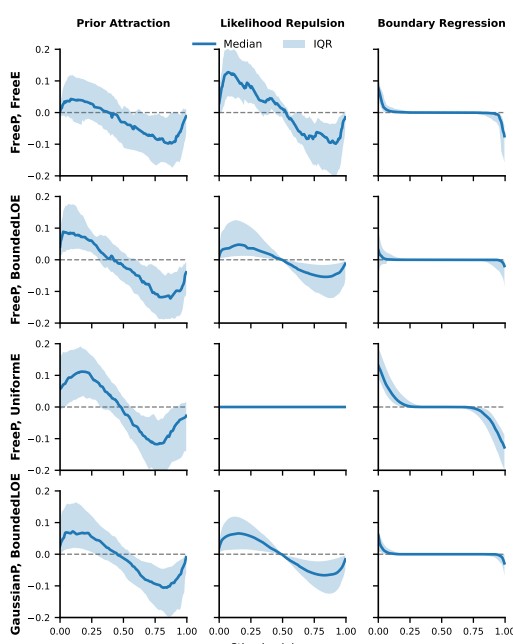

Figure 34: **DMR Pricing Task:** Bias decomposition of four Bayesian model variants in Figure 5B. The solid line shows the group median, and the shaded area indicates the interquartile range (IQR). Plotting details are provided in Appendix C.14.

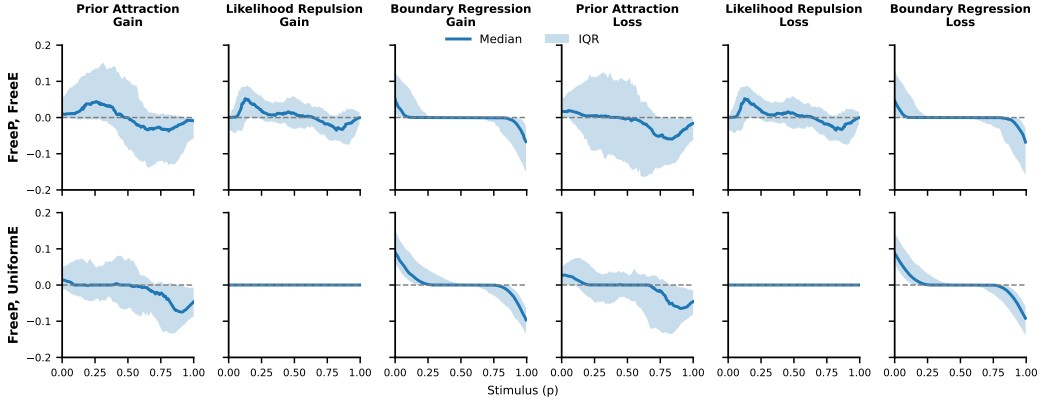

Figure 35: **DMR Choice Task:** Bias decomposition of two Bayesian model variants in Figure 5D. The solid line shows the group median, and the shaded area indicates the interquartile range (IQR). Plotting details are provided in Appendix C.14.

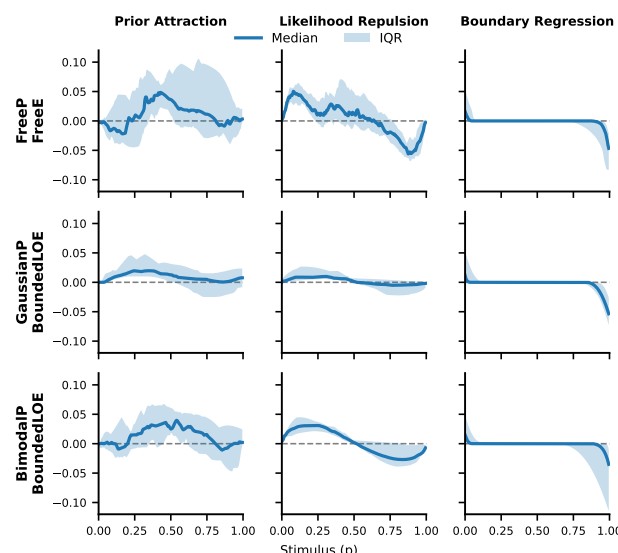

Figure 36: **Adaptation Task:** Bias decomposition of three Bayesian model variants in Figure 6D. The solid line shows the group median, and the shaded area indicates the interquartile range (IQR). Plotting details are provided in Appendix C.14.

### C.14 DETAILS FOR PLOTTING GROUP-LEVEL PRIOR AND RESOURCES

For each group-level curve presented in the paper, we plot the median across subjects as the solid line and the interquartile range (IQR; 25th–75th percentile) as the shaded area. These quantities are computed pointwise on the stimulus grid, providing a summary of central tendency and between-subject variability.

### C.15 METHODS FOR CALCULATING BIAS AND VARIANCE

This section describes the procedures used to calculate bias and variance for the non-parametric data, the Bayesian model, and the BLO model for the bias and variance figures (for example, Figure 14 and Figure 15).

#### C.15.1 NON-PARAMETRIC ESTIMATION

The non-parametric estimate of relative frequency is defined as

$$\hat{\pi}_{NP}(p) = \frac{1}{m} \sum_{t=1}^{m} \hat{\pi}_t(p),$$

where $\hat{\pi}_t(p)$ denotes the subject's estimate on trial $t$, $t = 1, 2, \ldots, m$.

**Bias.** Bias is the difference between the mean estimate and the true probability:
$$\text{Bias}_{NP}(p) = \hat{\pi}_{NP}(p) - p.$$

**Variance.** The variance is computed as the sample variance of $\hat{\pi}_t(p)$ across trials of the same $p$.

#### C.15.2 BLO MODEL

The BLO model produces deterministic predictions $\hat{\pi}_{BLO}(p, N)$ that depend on numerosity $N$. To obtain a single prediction per probability $p$, we average across the five numerosity conditions ($N = \{200, 300, 400, 500, 600\}$):

$$\mathbb{E}[\hat{p}]_{\text{BLO}} = \frac{1}{5} \sum_{N \in \{200, 300, 400, 500, 600\}} \hat{\pi}_{\text{BLO}}(p, N).$$

**Bias.** Bias is then defined as

$$\text{Bias}_{\text{BLO}}(p) = \mathbb{E}[\hat{p}]_{\text{BLO}} - p.$$

**Variance.** Variance in BLO arises from Gaussian noise in log-odds space, $\epsilon_\lambda \sim \mathcal{N}(0, \sigma_\lambda^2)$. Mapping back into probability space requires a Jacobian transformation:

$$f(p) = \frac{1}{p(1-p)} \cdot \frac{1}{\sqrt{2\pi\sigma_\lambda^2}} \exp\left(-\frac{(\lambda(p) - \hat{\Lambda}_\omega)^2}{2\sigma_\lambda^2}\right).$$

The probability distribution is normalized, and the variance is computed as

$$\text{Var}_{\text{BLO}}(p) = \mathbb{E}[P^2] - \left(\mathbb{E}[P]\right)^2,$$

where expectations are taken with respect to $f(p)$.

### C.15.3 BAYESIAN MODEL

For the Bayesian model, the estimate given a measurement $m$ is $\hat{p}(m)$. The mean estimate for stimulus $p$ is

$$\hat{p}(p) = \sum_m \hat{p}(m)\, P(m \mid p).$$

**Bias.** Bias is defined as

$$\text{Bias}_{\text{Bayesian}}(p) = \hat{p}(p) - p.$$

**Variance.** Variance has two components:

$$\text{Var}_{\text{Bayesian}}(\hat{p}(m) \mid p) = \underbrace{\sum_m P(m \mid p)\left(\hat{p}(m) - \hat{p}(p)\right)^2}_{\text{sensory variance}} + \underbrace{\sigma_{\text{motor}}^2}_{\text{motor variance}}.$$

### C.16 EXPERIMENTS ON EXPLAINING PROBABILITY DISTORTION IN VISION-LANGUAGE MODELS

We evaluate two open-source vision–language models (VLMs)—InternVL3-14B and Qwen-VL2.5-7B—on a judgment-of-relative-frequency task. Both models exhibit the inversed S-shaped probability distortion commonly observed in humans. Probing hidden representations further reveals a U-shaped profile of discriminability (used as a proxy for Fisher information, FI), consistent with a Bayesian account in which boundary regions ($p = 0$ or $p = 1$) are encoded with higher precision.

#### C.16.1 TASK AND STIMULI

**Visual stimuli.** We generated dot-array images with total counts $N \in \{200, 300, 400, 500, 600\}$ and black-dot proportions $p \in \{0, 0.01, 0.02, \ldots, 0.99, 1.0\}$. For each $(p, N)$ combination, 40 images were created, yielding 20,200 images in total.

**Text-only stimuli.** To test whether the observed distortions depended on vision, we also converted dot arrays into textual descriptions. Each image was replaced by a string of Unicode characters (e.g., black and white circles) preserving the same proportions. This allows us to present the same task in a purely language-based format.

**Prompting.** Each image was presented to a VLM with two types of prompts: a long descriptive instruction asking for a careful estimate and a shorter instruction.

- **Prompt 1:** *"This image shows a cluster of black and white dots. Without needing an exact count, can you quickly visually estimate in 1 second what percentage of the dots are black? Please estimate as precisely as possible the percentage of white dots among all the dots in the image. Provide only a number between 1 and 100, and you can include decimals."*
- **Prompt 2:** textit*"Estimate the proportion of black dots, give only a number between 1 and 100."*

**Model Responses Processing.** Numeric outputs were parsed and normalized to $[0, 1]$. All outputs fell within the instructed range of 1–100.

### C.16.2 BEHAVIORAL FINDINGS: S-SHAPED DISTORTION

We first examined the end-to-end behavior of open-source VLMs when prompted with dot-array images. Both InternVL3-14B and Qwen-VL2.5-7B produced probability estimates showing the classic S-shaped distortion observed in humans (Figures 37, 38). Specifically, models overestimated small probabilities, underestimated large probabilities, and showed a dip in accuracy near $p = 0.5$, accompanied by relatively low variability—a pattern similar to human data. These results held across both long and short prompts (Figures 39, 40).

To test whether the S-shaped bias depends on visual processing, we presented models with text-only stimuli, where dot arrays were converted into Unicode-based descriptions. In this condition, the S-shape disappeared: responses were approximately linear with a monotonic error increase (Figure 41). Similar patterns were observed across models and prompts, suggesting that the bias is not intrinsic to language-only reasoning. For comparison, we also trained simple vision-only baselines (a two-layer MLP and a CNN) on dot images with regression and classification objectives. These controls suggest that the S-shape emerges specifically from joint visual–textual processing.

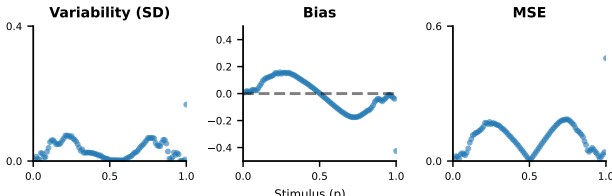

Figure 37: InternVL3-14B (Prompt 1): variability (SD), bias, and mean squared error (MSE) with respect to $p$. The bias curve is S-shaped with a variability dip near $p = 0.5$.

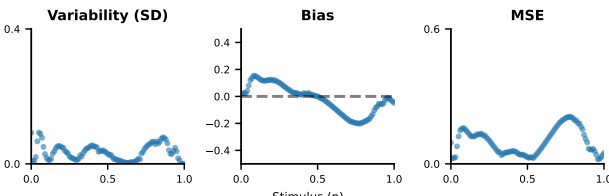

Figure 38: Qwen-VL2.5-7B (Prompt 1): S-shaped bias and a variability dip near $p = 0.5$.

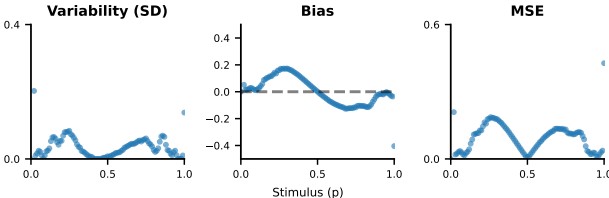

Figure 39: InternVL3-14B (Prompt 2): similar pattern to prompt 1 (Figure 37).

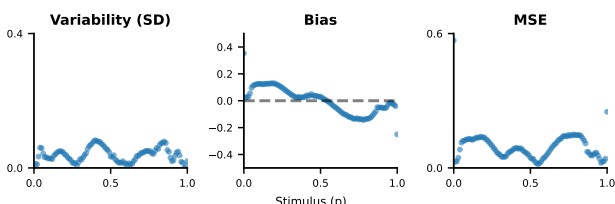

Figure 40: Qwen-VL2.5-7B (Prompt 2): similar pattern to Prompt 1 (Figure 38).

For comparison, we trained two simple baselines (a two-layer MLP and a CNN) on the same images with regression and classification objectives. Neither reproduced the S-shaped bias. We also tested a text-only prompt condition by converting dot images into Unicode-based textual descriptions. In this setting, the S-shape disappeared: the model showed nearly linear responses, with systematic underestimation at higher probabilities and a monotonic increase in error (QwenVL2.5-7B, prompt 1; Figure 41). The results of InternVL model and the other prompts show highly similar linear bias as well. This suggests that the model tend to default to a narrow range of answers regardless of the input. Taken together, these results indicate that the S-shaped distortion emerges specifically from joint visual–textual processing rather than vision or text alone.

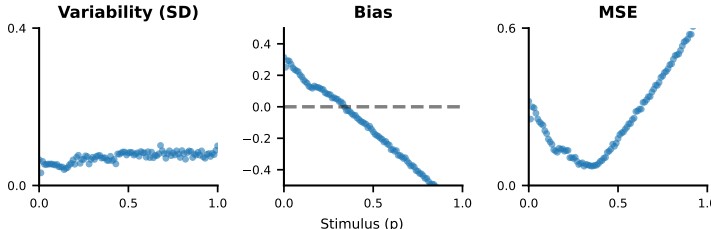

Figure 41: Qwen-VL2.5-7B, text-only (Prompt 1 on Unicode dot strings). The S-shaped bias disappears, replaced by a roughly linear bias.

### C.16.3    REPRESENTATION ANALYSIS WITH A PROXY FOR FISHER INFORMATION

We next asked whether the observed bias could be linked to how dot proportions are encoded internally. To quantify encoding resources, we measured discriminability, or Fisher information—the ability to distinguish a given proportion from nearby values. Operationally, we trained logistic regression classifiers to discriminate between adjacent proportions $(p_k, p_{k+1})$ based on hidden representations, and computed AUC scores on test sets. Plotting AUC against the pair midpoint $m_k = (p_k + p_{k+1})/2$ yields a curve $\text{AUC}_\ell(m_k)$, which we treat as a proxy for Fisher information.

We applied this method to hidden states in the decoder part of each VLM, where vision and text features interact. Both InternVL3-14B and Qwen-VL2.5-7B exhibited robust U-shaped Fisher information profiles (Figures 42, 43). The U-shape was evident from the earliest fusion layers and remained stable through the final layers, with higher discriminability near $p = 0$ and $p = 1$. This pattern parallels our theoretical prediction that a U-shaped allocation of resources underlies S-shaped bias.

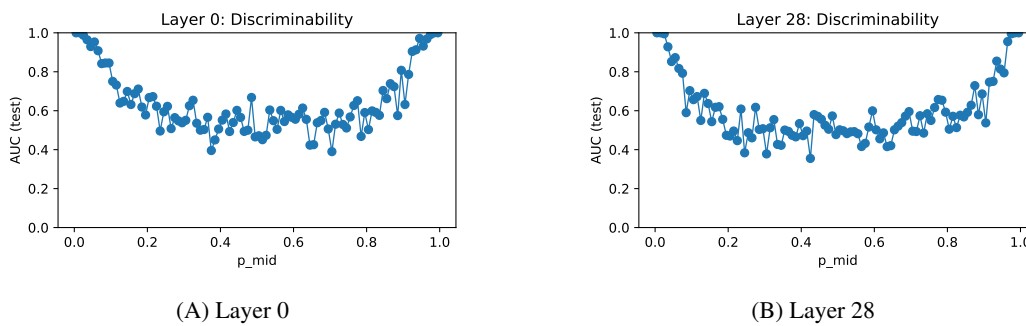

(A) Layer 0              (B) Layer 28

Figure 42: Qwen-VL2.5-7B: discriminability (AUC) as a proxy for FI. Both early and late layers are U-shaped.

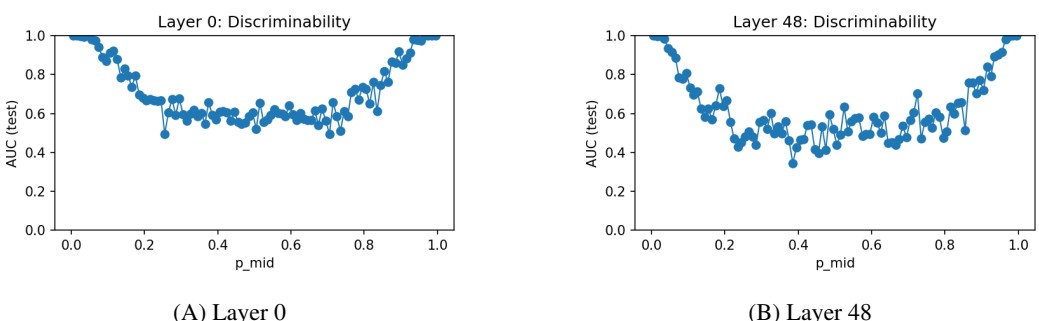

(A) Layer 0              (B) Layer 48

Figure 43: InternVL3-14B: discriminability (AUC) is likewise U-shaped.

### C.16.4 SUMMARY AND THEORETICAL IMPLICATIONS

Together, these results show that S-shaped probability distortion also emerges in large VLMs when performing the JRF task.

This phenomenon aligns with the framework of Stewart et al. (2006), which demonstrates that naturally occurring probability expressions in language corpora follow a distribution clustered at extremes rather than a uniform distribution. Since VLMs are trained on vast datasets reflecting these ecological statistics, they likely internalize this extreme-heavy prior. This interpretation is further supported by recent findings from Zhu et al. (2025), who explicitly showed that LLMs exhibit probability weighting patterns precisely when pretrained on data following such ecological distributions.

In our experiments, probing hidden representations reveals a U-shaped Fisher Information, consistent with the Bayesian account that nonuniform allocation of encoding resources underlies the observed S-shaped bias.

