# OpenReview forum: "The Bayesian Origin of the Probability Weighting Function in Human Representation of Probabilities"
_ICLR.cc/2026/Conference — Submitted to ICLR 2026_

### Official Review · Reviewer_LsJ6 · 2025-10-26

**Soundness:** 3
**Presentation:** 2
**Contribution:** 2
**Rating:** 4
**Confidence:** 4

**Summary:**

This paper proposes a novel account for the origin of the inverse S-shaped probability weighting function, framing it as a consequence of optimal Bayesian decoding from a noisy internal representation. Within this encoding-decoding framework, a true probability $p$ is encoded into a noisy measurement $m$, which is then decoded to produce an estimate $\hat{p}$. The authors demonstrate that this model provides a superior quantitative fit by reanalyzing human data across multiple domains, including perceptual judgment, economic pricing, and risky choice.

**Strengths:**

The paper is clearly written. It is supported by a rigorous theoretical and mathematical analysis of the Bayesian encoding-decoding framework for probabilities. Moreover, the authors provide compelling empirical support by reanalyzing a range of existing datasets and models, demonstrating that their new model achieves a strong quantitative fit.

**Weaknesses:**

My main comment is about the parameterization of the FreeP and FreeE in the Bayesian model. The non-parametric specification of the prior and encoding function, with 200 free parameters each, grants the model significant flexibility. While the authors note the difficulty in penalizing this complexity, this remains a critical point for model comparison, and a more robust treatment of this issue would strengthen the paper.

A related work also found that LLM embeddings exhibit patterns from probability weighting function when the LLM is pretrained on ecological data of probabilities (i.e., those following a bimodal distribution). See Figure 2 of Zhu, Yan, & Griffiths (2025). Similarly, an alternative Bayesian encoding-decoding may be developed when the $P_{prior}(p)$ is a bimodal distribution. There are some evidence suggesting that the bimodal distribution of subjective probabilities, for example, in English (Stewart et al., 2006).

Minor comment:

Line 1133: "$\Delta AICc$" --> "$\Delta AICs$"?

References:

Zhu, Yan, & Griffiths (2025) Language Models Trained to Do Arithmetic Predict Human Risky and Intertemporal Choice. ICLR.

Stewart, N., Chater, N., & Brown, G. D. (2006). Decision by sampling. Cognitive psychology, 53(1), 1-26.

**Questions:**

The authors briefly mentioned a VLM experiment in the last bit of the Appendix. It shows similar human-like distortion in probability estimation from VLMs. This is an interesting result, and probably way beyond the scope of the existing work. What is the main motivation for running this experiment?

---

> ### Author Response · Authors · 2025-11-22
>
> We thank for the reviewer for the close reading and for the feedback, which has helped us improve the paper.
>
> ### Regarding Weaknesses:
>
> > My main comment is about the parameterization of the FreeP and FreeE in the Bayesian model. The non-parametric specification of the prior and encoding function, with 200 free parameters each, grants the model significant flexibility. While the authors note the difficulty in penalizing this complexity, this remains a critical point for model comparison, and a more robust treatment of this issue would strengthen the paper.
>
>
>
> We agree that model complexity is crucial in comparison.
>
> First, we believe that **our primary metric heldout NLL is a very robust method** for guarding against overfitting.
>
> Second, to add more transparency, we computed an upper bound AIC using the full grid size as the parameter count (e.g., $k=403$ for the JRF task, 200 for encoding, 200 for prior, 3 for noise/mixing parameters). Note that this is very conservative (i.e., biased against ninparametric models), because the true number of degrees of freedom is substantially lower due to the smoothness of the fitted function. Nonetheless, we found that **the freely fitted Bayesian model still yields the lowest Summed $\Delta$AICc** with this conservative penalty. This confirms that the nonparametric models fit the data best even in an evaluation biased against them. We have updated the figure captions in the Appendix (JRF: line 1444, DMR Pricing: line 1745, DMR Choice: line 1890, Adapatation: line 2047) to report these AICc values.
>
>
> > A related work also found that LLM embeddings exhibit patterns from probability weighting function when the LLM is pretrained on ecological data of probabilities (i.e., those following a bimodal distribution). See Figure 2 of Zhu, Yan, & Griffiths (2025). Similarly, an alternative Bayesian encoding-decoding may be developed when the P(prior) is a bimodal distribution. There are some evidence suggesting that the bimodal distribution of subjective probabilities, for example, in English (Stewart et al., 2006).
>
> We thank the reviewer for pointing out these interesting connections. We have incorporated a discussion of Zhu et al. (2025) and Stewart et al. (2006) in Appendix Section C.16.4 (line 2679 to line 2684), noting how ecological statistics may shape internal representations.
>
> That said, our formal model comparison unambiguously establishes that the probability distortion observed in our specific datasets is primarily driven by **nonuniform encoding resources** (Likelihood Repulsion) rather than attraction to a bimodal prior. As shown in our results (e.g., Figure 3A), a nonuniform prior alone is insufficient to capture the full behavioral pattern without the corresponding U-shaped encoding resources.
>
>
> ### Regarding Minor Comment:
>
> >Line 1133:
> "dAICc--> dAICs"?
>
> We confirm that this refers to the difference in the Corrected Akaike Information Criterion (AICc) with a correction for small sample sizes.
>
>
>
> ### Regarding Questions:
> > The authors briefly mentioned a VLM experiment in the last bit of the Appendix. It shows similar human-like distortion in probability estimation from VLMs. This is an interesting result, and probably way beyond the scope of the existing work. What is the main motivation for running this experiment?
>
> Our motivation was to understand if nonuniform FI also drives biases in artifical systems. Finding similar patterns in VLMs suggests that such distortions may arise from the statistical structure of training data. Moreover, since artificial models allow for direct inspection of internal representations, this experiment provided an opportunity to directly infer the shape of the encoding resource allocation.

---

> ### Author Response · Authors · 2025-11-28
>
> Dear Reviewer,
>
> Thanks again for your feedback. We really appreciate your time and effort!
>
> As described above, we have addressed your concerns in the revision, particularly regarding the analysis of model complexity (AIC) and discussion of further related work.
>
> We'd be grateful if you could let us know if you have any further questions. If you consider our response satisfactory, we'd also appreciate if you could update your score accordingly.
>
> Best regards,
> Authors

---

### Official Review · Reviewer_KhJE · 2025-10-28

**Soundness:** 3
**Presentation:** 3
**Contribution:** 2
**Rating:** 4
**Confidence:** 4

**Summary:**

This submission proposes an noisy encoding+Bayesian decoding framework as a way of providing a general account for well documented biases in the ability to estimate probabilities in humans. The framework generalizes on a range of recent encoding-decoding models (which can be seen as special cases with the framework) arguing for a few discriminatory signatures in terms of the fisher information of the encoding procedure (U-shaped, usually) and in behavior (bias and across trial variability). The model is tested on several public dataset and a small new human dataset collected specifically to validate model predictions. For these, it shows improvements in model quality against a few chosen alternatives from the literature.

**Strengths:**

The ability to put together a collection of related models from the literature under a single framing helps the cogsci community make sense of the subfield.

Decomposing the bias into individual contribution from encoding (resource allocation) and decoding (prior) allows for making new predictions about changes in prior, in particular bi-modality, which were then validated using data from a purposefully designed new experiment.

**Weaknesses:**

Clarity of core theoretical results description: The main problem for me was the description of the core theoretical results in section 3. Theorem 1 is opaque in the main text and the proof in the appendix is incomprehensible.  The links between the actual math and the predictions is not spelled out clearly enough to drive the point across that the theory is instrumental in driving the rest of the analysis. The core theoretical result should be understandable to a mathematically educated reader based on the paper text alone and this is very much not happening at the moment.

Novelty: The theoretical framing seems to rely heavily on past bayesian accounts of estimation biases in other perceptual domains, both in conceptualization and in the math (and the authors do reference the links to this work explicitly). It is not clear to me what aspects of the derivation are unique to this specific problem setting rather than  an extrapolation of a well understood broader phenomenon. does the bounded nature of the message being conveyed change anything fundamentally ? does it require a different technical approach to the derivation? It seems that the properties of the Fiser information at the limit were inherited from derivations elsewhere, and outside of the bounds the general framing applies... The variance results seem lifted directly from Hahn et al. (2025) with no added value. But I could be missing something since the appendix derivations are missing critical bits of context and explanation and are largely impossible to follow in the details.  In any case, the text itself should make a stronger case for what exactly is technically and conceptually novel in this particular version, in particular when compared to recent Hahn et al 2024,2025.

Comparison to existing models: the introduction mentions a wide range of models, yet BLO is almost invariably used as the straw-man for data comparisons. While it is not reasonable to ask comparisons to many other models at the very minimum one would expect a justification for why the authors deemed this particular model to be the most important to compare to and some thoughts about what one would expect for other models. The other concern I have is about the model complexity correction treatment: it seems like the nonparametric models behave fundamentally differently in terms of the metrics used and they do not get penalized in any way for the additional degrees of freedom. again, i don't necessarily argue for changing the metric but i think the text needs to be more transparent about the implications of the metrics used wrt nonparametric models, just saying that it's hard to do an AIC equivalent for that model class in the appendix doesn't seem sufficient.

The general conceptualization of the problem is widely used in this community and not due to the authors, but to me is somewhat suspect as a premise; this is a personal opinion and does not affect my scores in any way but i am generally dubious on the idea of treating probability of a world event the same way one would any sensory latent. What the empirical data looks like to me is a general flattening of the agent's posterior beliefs about the latent variable being judged (whose probability is being manipulated) relative to an ideal observer with perfect understanding of the task and no additional sensory uncertainty. So yes, it does seem normatively sensible to be overall more uncertain about what is going on and that then affecting the decision variable resulting in a combination of bias and variance. It is just not clear to me why invoking an arbitrary F nonlinearity (and associated Fiser information) and fitting the data in this parametrization really says anything useful about the mechanistic nature of the process or provide additional insight rather than being a more fancily motivated variation of the descriptive models of old.

Model fitting procedures are described in a very limited way and seem to rely heavily of a public library, enforcing the incremental feel of the work.

**Questions:**

Figure 3: why not show both uniform and U-shaped scenarios in B and C? i think the paper could benefit from helping the reader build more intuition about the tradeoffs that different model choices make in terms of different contributions to the bias (this also applied to the associated main text which could use some expansion and editing for clarity )

Figure 3 A: there seems to be a tradeoff between capturing bias and variance patterns well in the human data, any intuition for why ?

Figure 5: i am confused why the uniform model shows such a poor model comparison result when the inferred fisher information for the nonparametric model is the flattest of all dataset. more generally, there seems to be a not very intuitive relationship between the metric and the ability of the models to capture qualitative features in the data in the sense defined by the theory, which may need some discussion.

Figure 7: BLO seems to also imply a U shaped fisher information curve, so it's not clear to me if the predictions of the model are generally that qualitatively different from past variants. It seems like differences in variability is where the model improvement quality is coming from. Please comment.

---

> ### Author Response · Authors · 2025-11-21
> **Response (Part 1)**
>
> We thank the reviewer for the close reading and for the feedback, which has helped us improve the paper.
>
> ### Regarding Weaknesses:
>
> > Clarity of core theoretical results description: The main problem for me was the description of the core theoretical results in section 3. Theorem 1 is opaque in the main text and the proof in the appendix is incomprehensible. The links between the actual math and the predictions is not spelled out clearly enough to drive the point across that the theory is instrumental in driving the rest of the analysis. The core theoretical result should be understandable to a mathematically educated reader based on the paper text alone and this is very much not happening at the moment.
>
> We thank the reviewer for this feedback. We have expanded and rephrased this section (from line 153 to line 195), especially Theorem 1, to improve clarity. We have also restructured the proof in Appendix B.1.1 (from line 806 to line 920) to be more step-by-step and readable.
>
> > Novelty: The theoretical framing seems to rely heavily on past bayesian accounts of estimation biases in other perceptual domains, both in conceptualization and in the math (and the authors do reference the links to this work explicitly). It is not clear to me what aspects of the derivation are unique to this specific problem setting rather than an extrapolation of a well understood broader phenomenon. does the bounded nature of the message being conveyed change anything fundamentally ? does it require a different technical approach to the derivation? It seems that the properties of the Fiser information at the limit were inherited from derivations elsewhere, and outside of the bounds the general framing applies... The variance results seem lifted directly from Hahn et al. (2025) with no added value. But I could be missing something since the appendix derivations are missing critical bits of context and explanation and are largely impossible to follow in the details. In any case, the text itself should make a stronger case for what exactly is technically and conceptually novel in this particular version, in particular when compared to recent Hahn et al 2024,2025.
>
>
> First, yes, the bounded nature of probability requires new technical  contributions. We have considerably reworked the proof of Theorem 1 in Appendix B.1.1 to clarify which aspects are novel, and how this result goes above and beyond Hahn et al 2024, 2025. In particular, prior results do not consider the setting of a stimulus which has two boundaries (upper and lower); accounting for this is key to Theorem 1.
>
> Second, we would like to remark that theoretical understanding of BLO and clarifying its link to Bayesian models is a substantive contribution, which had not been attempted in prior work. We consider Theorem 2 and the further results shown in B.1.2 (referenced in prose in the main paper) to be substantive theoretical contributions.
>
>
> > Comparison to existing models: the introduction mentions a wide range of models, yet BLO is almost invariably used as the straw-man for data comparisons. While it is not reasonable to ask comparisons to many other models at the very minimum one would expect a justification for why the authors deemed this particular model to be the most important to compare to and some thoughts about what one would expect for other models.
>
> We would like to clarify that we compare against all three  relevant groups of previous models, but we understand our writing inadvertently hid some of this comparison:
> * traditional prospect theory with fixed probability distortion functions, still by far the dominant framework. Zhang et al. (2020) already showed that BLO outperforms such models on the JRF and Pricing datasets, which is why we don't include these models there. We do, however, explicitly compare on the Choice dataset.
> * prior Bayesian models: These have never been studied in either theory or modeling at nearly the breadth we're doing. They can be viewed as special cases of our framework and we in fact continuously compare with them. E.g. Fennell & Baddeley (2012) and Bedi et al. (2025) correspond to the case where resources are uniform and distortion is attributed to boundary regression, Frydman & Jin (2023) correspond to the case where FI is proportional to the prior. Our results show that such models underperform our proposed model (U-shaped FI, task-dependent prior).
> * BLO: the most thoroughly evaluated non-traditional model; we compare to it in JRF and Pricing datasets, as it's state-of-the-art on those.
>
> We have rewritten to make transparent that we in fact compare against all these previous models; please see the changes (in blue) throughout the paper. Thanks for this feedback, which helped us improve the presentation.

---

> ### Author Response · Authors · 2025-11-22
> **Response (Part 2)**
>
> > The other concern I have is about the model complexity correction treatment: it seems like the nonparametric models behave fundamentally differently in terms of the metrics used and they do not get penalized in any way for the additional degrees of freedom. again, i don't necessarily argue for changing the metric but i think the text needs to be more transparent about the implications of the metrics used wrt nonparametric models, just saying that it's hard to do an AIC equivalent for that model class in the appendix doesn't seem sufficient.
>
> We agree that model complexity is crucial in comparison. We would like to remark that our primary metric Negative Log-Likelihood (NLL) on held-out data guards against overfitting. To provide full transparency, we computed a upper bound AIC using the full grid size as the parameter count (e.g., $k=403$ for the JRF task, 200 for encoding, 200 for prior, 3 for noise parameters). We note that this is extremely conservative (i.e., biased against the nonparameteric model), as the true effective parameter count of the model is much smaller due to the smoothness of the fitted functions. Nonetheless, even then we found that the freely fitted Bayesian model **still yields the lowest Summed $\Delta$AICc** with this very conservative penalty.  We have updated the figure captions in the Appendix (JRF: line 1444, DMR Pricing: line 1745, DMR Choice: line 1890, Adapatation: line 2047) to report these results.
>
>
> > The general conceptualization of the problem is widely used in this community and not due to the authors, but to me is somewhat suspect as a premise; this is a personal opinion and does not affect my scores in any way but i am generally dubious on the idea of treating probability of a world event the same way one would any sensory latent. What the empirical data looks like to me is a general flattening of the agent's posterior beliefs about the latent variable being judged (whose probability is being manipulated) relative to an ideal observer with perfect understanding of the task and no additional sensory uncertainty. So yes, it does seem normatively sensible to be overall more uncertain about what is going on and that then affecting the decision variable resulting in a combination of bias and variance. It is just not clear to me why invoking an arbitrary F nonlinearity (and associated Fiser information) and fitting the data in this parametrization really says anything useful about the mechanistic nature of the process or provide additional insight rather than being a more fancily motivated variation of the descriptive models of old.
>
> We agree that treating abstract probability as a sensory latent is a nontrivial assumption. However, we argue that our framework can be used to unify disparate behaviour phenomena (bias, variance and adapation) better than traditional models.
>
> **Regarding the mechanistic nature:**
>
> First, F is mechanistic. We do not arbitrarily choose F to be a probability weighting function. It is a (simplified) model of the neural code. We infer the specific shape of F from data and recover a map of neural resources, which tell us that the brain's representation of probability is optimized to distinguish certain probability regions better than others, so it is not uniform.
>
> Second, a standard weighting function (e.g. Prelec) is static, which our model could adapt to changes in stimulus distribution without changing the underlying encoding mechanism.
>
>
>
> **Why it is not a variation of old models:** BLO was an earlier attempt at providing an explanatory model and it relied on a heuristic (truncation and scaling). We argue that the Bayesian model supersedes that, because it shows that BLO is an approximation of the optimal Bayesian account. By removing BLO's arbitrary constraints, we show that human behaviour aligns with rational inference over U-shaped resources. The model parsimoniously explains the origin of probability distortion, including the effect of the prior.
>
> > Model fitting procedures are described in a very limited way and seem to rely heavily of a public library, enforcing the incremental feel of the work.
>
> The main technical contribution of the present paper is the **theoretical and conceptual analysis** of the Bayesian model family and its relation both to traditional probability distortion and BLO, together with **extensive evaluation** on behavioral data. It is true that we use a public library for implementing Bayesian models; we consider this a strength, as the library of Hahn et al 2024, 2025 has already been thoroughly validated in prior work on other stimulus domains.

---

> ### Author Response · Authors · 2025-11-22
> **Response (Part 3)**
>
> ### Regarding Questions
>
> > Figure 3: why not show both uniform and U-shaped scenarios in B and C? i think the paper could benefit from helping the reader build more intuition about the tradeoffs that different model choices make in terms of different contributions to the bias (this also applied to the associated main text which could use some expansion and editing for clarity )
>
> Thank you for this insight. To build more intuition, we have added a bias decomposition figure in the Appendix Figure 33 (line 2351). This figure shows the trade-offs in model choices:
>
>
> **Uniform Encoding (Row 3):** As shown in the Likelihood Repulsion column, a uniform encoding produces zero repulsion bias. Consequently, this model relies entirely on Boundary Regression and Prior Attraction. This explains the unnatural fit observed in the main results. Without the gradual change in precision provided by Likelihood Repulsion. This results in the bias curve that is rigidly constrained at the boundaries and overly reliant on the prior in the center.
>
> **U-shaped Encoding (Rows 1, 2, 4):** In contrast, models with U-shaped FI generate a S-shaped Likelihood Repulsion across the entire range. This component is essential for reproducing the characteristic human bias pattern, confirming our theoretical prediction that U-shaped resources are necessary for S-shaped weighting.
>
> We have also added figure 4 (line 378) containing uniform and freely fitted encoding and discussed it in Section 4.1 (line 355 to line 359).
>
> > Figure 3 A: there seems to be a tradeoff between capturing bias and variance patterns well in the human data, any intuition for why ?
>
> The trade-off could be observed in two places:
> Model variants with Bounded Log-odds encoding fit the bias the best (even better than the model variants with freely fitted encoding), while they fail to capture the variance pattern.
>
> The intuition for these phenomenon is:
>
> The prior probability mass sums to 1; to capture the sharp dip in variability at $p=0.5$, the FreeP model allocates a large amount of probability mass to the center (creating a sharp peak in Figure 12). This strong attraction distorts the shape of the prior at the boundaries. The optimization  accepts the resulting slight distortions in the bias curve (particularly in the high-variance regions) to prioritize fitting the variance dip at the center.
>
> In contrast, the GaussianP model is constrained by parametric rigidity. It cannot decouple its central peak from its global width. To fit the global S-shaped bias, it must fit a relatively larger $\sigma$, which prevents it from forming the sharp peak. Thus, it sacrifices the variance fit to maintain a better global bias fit.
>
>
>
> > Figure 5: i am confused why the uniform model shows such a poor model comparison result when the inferred fisher information for the nonparametric model is the flattest of all dataset. more generally, there seems to be a not very intuitive relationship between the metric and the ability of the models to capture qualitative features in the data in the sense defined by the theory, which may need some discussion.
>
> We have added bias decomposition plots (Figure 35, line 2414) for these models in the DMR Choice Task to better illustrate the mechanism.
>
> Across all tasks, precision near the endpoints ( $p=0$ and $p=1$ ) is critical, but it is particularly vital in DMR Choice Task where discriminability drives preference. To consistently choose one option over another, the observer must be able to clearly distinguish between probabilities.
>
> As the decomposition shows, the UniformE model has zero Likelihood Repulsion, and Prior Attraction fails to compensate for this deficit. This is likely because prior cannot restore the lost sensory discriminability and consequently  the optimizer defaults to a flatter prior to avoid introducing additional bias error.
>
> Quantitatively, the UniformE model performs poorly in the JRF task as well. While its fit is marginally better in the DMR Pricing task, this is likely because Pricing requires generating a value (Certainty Equivalent) rather than making a binary discrimination, allowing the prior to partially mask the encoding limitations.
>
> That said, we would like to remark that our results confirm that U-shaped encoding and the resulting Likelihood Repulsion is a key component for recovering the human bias.

---

> ### Author Response · Authors · 2025-11-22
> **Response (Part 4)**
>
> > Figure 7: BLO seems to also imply a U shaped fisher information curve, so it's not clear to me if the predictions of the model are generally that qualitatively different from past variants. It seems like differences in variability is where the model improvement quality is coming from. Please comment.
>
> Yes, the BLO model also implies a U-shaped fisher information. However, there are three fundamental differences between the FI of BLO model and the freely fitted Bayesian model:
>
> 1. BLO **enforces** the U-shape via a parametric assumption (truncated log-odds), which introduces **discontinuities**. We have also proven this property in Theorem 8 and visualized it in Figure 11. These discontinuities are not grounded on human data. In contrast, our framework **recovers** U-shaped resources nonparametrically from the data. This validates that the U-shape is an intrinsic property of human probability representation, without starting from a parametric assumption.
> 2. While bias predictions are not qualitatively different for standard tasks (JRF/DMR Pricing), they diverge in Adaptation Task. Because BLO relies on a fixed transformation, it predicts an invariant S-shaped bias regardless of stimulus statistics. In the Adaptation Task, however, human bias becomes multi-peaked to reflect the bimodal stimuli distribution. BLO fails to capture this bias pattern.
> 3. Indeed, improvement comes from better fit to the variability. We would like to argue that this is theoretically critical, as the ability to better capture the overall response distribution -- not just the mean response (bias) -- shows the superiority of Bayesian decoding at modeling the human behavior overall.

---

> ### Author Response · Authors · 2025-11-28
>
> Dear Reviewer,
>
> Thanks again for your feedback. We really appreciate your time and effort!
>
> As described above, we have addressed your concerns in the revision, particularly regarding the clarification of our theoretical contributions, the rigor of model comparison, and the intuition behind the results.
>
> We'd be grateful if you could let us know if you have any further questions. If you consider our response satisfactory, we'd also appreciate if you could update your score accordingly.
>
>
> Best regards,
>
> Authors

---

### Official Review · Reviewer_WQML · 2025-10-31

**Soundness:** 3
**Presentation:** 1
**Contribution:** 2
**Rating:** 4
**Confidence:** 3

**Summary:**

This paper presents a Bayesian framework for modelling representation of probability in the brain. It accounts for different biases that have been identified and modeled in the literature. The proposed framework is optimal under the assumption that the internal representation of probability distribution is not perfect (there is a noise in this representation). Tested on different tasks that require probability estimation, the authors show that an optimal Bayesian model with imperfect representation of probability outperforms a method based on sub-optimallity (bounded rationality) assumption. While the results are interesting, I had a hard time understanding the contribution of this work in terms of modelling and for the ICLR audience.

**Strengths:**

The paper/presented model performed a good job in incorporating different models and biases. The diversity of experiments was also impressive. Furthermore, the proofs/theories were good contributions.

**Weaknesses:**

This might be a presentation issue, but in general I struggled with understanding originality/contribution of the paper specially in terms of modelling. It looks like to me that the framework is the sum of previous models already existed in the literature. The results also suggested that not all of the components are always needed and of course different tasks produces different challenges and biases. Overall, I am not sure about the message of the paper. Is it "the Bayesian brain"? If that's the case I don't think showing one non-optimal model, i.e. BLO doesn't work on a few tasks is that interesting for the community.

**Questions:**

Please see Weaknesses section. Also:
- In the result section likelihood of held out data was used. It looks like the number of free parameters was not considered. What do the results look like considering the complexity of models and their free parameters (e.g. by measures like AIC)
- What is the contribution of this paper beyond collecting the existing biases/models and/or papers like (Hahn, Wei 2024)?

---

> ### Author Response · Authors · 2025-11-21
>
> We thank the reviewer for the close reading and for the feedback, which has helped us improve the paper, especially the presentation of our novel contributions.
>
> ### Regarding Weaknesses:
>
> > This might be a presentation issue, but in general I struggled with understanding originality/contribution of the paper specially in terms of modelling. It looks like to me that the framework is the sum of previous models already existed in the literature. The results also suggested that not all of the components are always needed and of course different tasks produces different challenges and biases. Overall, I am not sure about the message of the paper. Is it "the Bayesian brain"? If that's the case I don't think showing one non-optimal model, i.e. BLO doesn't work on a few tasks is that interesting for the community.
>
> The message to the community is: For decades, the field has stipulated a probability weighting function.
> Our paper is the **first to rigorously establish in theory and modeling** that Bayesian inference over imprecise representations more accurately accounts for human representation of probability.
>
> We do acknowledge prior work had articulated related ideas, and the presentation in the paper might have made look our contribution less innovative than it really is. In fact, no prior study has (i) attributed probability distortion to the likelihood repulsion component of the Bayesian estimator's bias, while (ii) demonstrating fit across estimation and choice datasets.
>
> There are three types of prior work to compare to:
> * traditional prospect theory, still by far the dominant framework.
> * prior Bayesian models: These have never been studied in either theory or modeling at nearly the breadth we're doing. They can be viewed as special cases of our framework.
> * BLO: the most thoroughly evaluated non-traditional model
>
> The reason we compare to BLO a lot is that the other explanatory models are all special cases of our modeling framework, and can be compared to within the Bayesian framework.
>
> We have edited the paper to bring out the contribution more clearly; please see the changes (in blue) throughout the paper. We believe that this substantially increases the paper's clarity and impact.
>
> ### Regarding Questions:
> > In the result section likelihood of held out data was used. It looks like the number of free parameters was not considered. What do the results look like considering the complexity of models and their free parameters (e.g. by measures like AIC)
>
> Thank you for raising this point. We initially excluded the nonparametric Bayesian model from this metric due to the difficulty of defining its effective degrees of freedom given the regularisation and dependence of parameters.
>
> We computed an upper bound AIC using the full grid size as the parameter count (e.g., $k=403$ for the JRF task). We found that **even with this conservative penalty, the freely fitted Bayesian model yields the lowest Summed $\Delta$AICc**, confirming its superiority is not due to unpenalized flexibility. We have updated the figure captions in the Appendix (JRF: line 1444, DMR Pricing: line 1745, DMR Choice: line 1890, Adapatation: line 2047) to report these AIC values for transparency.
>
> > What is the contribution of this paper beyond collecting the existing biases/models and/or papers like (Hahn, Wei 2024)?
>
> Please see our response to the Weakness.

---

> ### Author Response · Authors · 2025-11-28
>
> Dear Reviewer,
>
> Thanks again for your feedback. We really appreciate your time and effort!
>
> As described above, we have addressed your concerns in the revision, particularly regarding the analysis of model complexity (AIC) and the clarification of our theoretical contributions.
>
> We'd be grateful if you could let us know if you have any further questions. If you consider our response satisfactory, we'd also appreciate if you could update your score accordingly.
>
> Best regards,
> Authors

---

### Official Review · Reviewer_poyN · 2025-11-01

**Soundness:** 3
**Presentation:** 3
**Contribution:** 3
**Rating:** 8
**Confidence:** 3

**Summary:**

The authors introduce a new Bayesian framework to explain how humans perceive and process probabilities. The authors claim that probability distortion is not deterministic, and that these distortions emerge naturally from how the brain encodes probabilities with noise and them optimally decodes them using Bayesian inference. Throughout their three experiments, the authors empirically demonstrate that their Bayesian framework consistently outperforms alternative models, including the current state-of-the-art BLO models. While the experiments have some limitations, particularly in sample size for the adaptation study and the use of hypothetical scenarios in economic tasks, the consistency of findings across different domains and tasks provides strong support for their theoretical framework. This work not only advances our understanding of how humans process probabilities but also demonstrates how fundamental principles of perception and inference can explain seemingly irrational aspects of decision-making.

**Strengths:**

The paper has strong theoretical contributions, backed with consistent empirical results and robust mathematical derivations. The authors provides a unifying Bayesian framework for understanding probability distortion in human cognition, derives clear theoretical predictions, and demonstrates how previous models can be seen as special cases. This theoretical unification is particularly powerful because it doesn't just describe probability distortions but explains why they occur, grounding them in fundamental principles of neural information processing. The framework is tested across multiple domains, showing consistent support for the key predictions laid out ahead of time. Most importantly, the mathematical derivations are sound and the experimental methodology is robust.

**Weaknesses:**

(See questions)

**Questions:**

1. The paper does not present power analysis to justify its samples sizes for different tasks. In particular, section 4.3's adaptation experiment only had 26 subjects, which roughly a third of what other experiments had (86 in 4.1 and 75 in 4.2). Is there a reason why there is a large difference in the number of subjects between experiments, and some support to show that 26 subjects was sufficient?

2. In 4.2, the decision-making tasks used were one-shot, hypothetical gambles without real consequences. While using actual monetary stakes is probably impractical, these hypothetical scenarios may not adequately capture how humans incorporate risk into their decision-making. In fact, it is a common criticism given to these one-shot, hypothetical gambling tasks often used in behavioral economics studies. Have authors considered using sequential decision-making tasks where time and physical effort could serve as natural proxies for risk and reward? For instance, in a task where completion speed determines when participants can leave, subjects would face real tradeoffs between speed and effort allocation.

3. While the Bayesian framework is more principled, it's also more complex. Could the authors comment on the practicality of using their framework compared to simpler alternatives that may do "well enough"?

---

> ### Author Response · Authors · 2025-11-23
>
> We thank the reviewer for the close reading and for the feedback.
>
> ### Regarding Questions:
> > The paper does not present power analysis to justify its samples sizes for different tasks. In particular, section 4.3's adaptation experiment only had 26 subjects, which roughly a third of what other experiments had (86 in 4.1 and 75 in 4.2). Is there a reason why there is a large difference in the number of subjects between experiments, and some support to show that 26 subjects was sufficient?
>
> We appreciate the suggestion to verify the robustness of our results. We performed a subject-level bootstrap analysis (10000 iterations) on the held-out NLL data to ensure our conclusions are not driven by outliers. That means, in each bootstrap replicate, we resampled 26 subjects from the original 26 subjects (with replacement). This allows us to check how robust our conclusions are to the 26 subjects in the experiment. As detailed in Appendix C.10.3 Table 2 (line 2071), the flexible Bayesian model significantly outperformed the Efficient-coding model (PriorMatchedE) in 98.1% of bootstrap samples (95% CI of NLL difference: [233, 7551]). Furthermore, it outperformed the BLO model in 100% of samples (95% CI: [7518, 13399]). This confirms that 26 subjects are sufficient to confidently establish the superiority of our Bayesian account in adaptation.
>
> > In 4.2, the decision-making tasks used were one-shot, hypothetical gambles without real consequences. While using actual monetary stakes is probably impractical, these hypothetical scenarios may not adequately capture how humans incorporate risk into their decision-making. In fact, it is a common criticism given to these one-shot, hypothetical gambling tasks often used in behavioral economics studies. Have authors considered using sequential decision-making tasks where time and physical effort could serve as natural proxies for risk and reward? For instance, in a task where completion speed determines when participants can leave, subjects would face real tradeoffs between speed and effort allocation.
>
> We agree that investigating risk in sequential decision-making is an interesting and exciting direction. However, we consider this out of the scope of our study, which focuses on understanding the specific mechanism of probability distortion. Given that prior work was prominently established and validated on standard one-shot discrete designs, using these same paradigms is necessary to perform a head-to-head model comparison without introducing  confounding factors in sequential tasks.
>
> > While the Bayesian framework is more principled, it's also more complex. Could the authors comment on the **practicality** of using their framework compared to simpler alternatives that may do "well enough"?
>
> We believe that Bayesian model is easy to use and does not require substantial computational resources. The non-parametric variant is most demanding, but it can still be run efficiently on a standard CPU. The model parameters are easy to fit on behavioral data as well.

---

> ### Author Response · Authors · 2025-11-28
>
> Dear Reviewer,
>
> Thanks again for your positive feedback on our theoretical and empirical contributions. We really appreciate your time and effort!
>
> We have posted a detailed response to your specific questions and made revisions accordingly. We would be grateful if you could let us know if these responses fully address your questions.
>
> We look forward to any further discussion.
>
> Best regards, Authors

---

### Author Response · Authors · 2025-12-03
**Global Response**

**Summary of Revisions and Responses**

We thank the reviewers for their constructive comments, which have helped us significantly strengthen the paper's theoretical clarity and empirical rigor. We have uploaded a revised PDF (changes in blue) and summarize our key responses below:

> What is the contribution of this paper beyond existing models (e.g., BLO) or frameworks?

While prior work proposed descriptive methods and speific instances of Bayesian inference, our paper is the first to rigorously establish, through both theory and empirical results, that Bayesian inference over imprecise representations provides a unified account of probability distortion across estimation and choice.

Beyond BLO: We clarified that BLO is treated as a descriptive parametric baseline. Our contribution lies in testing theoretical hypotheses. We now explicitly compare against traditional prospect theory, Uniform Encoding, and Efficient Coding accounts, demonstrating that only our flexible Bayesian framework unifies the S-shaped bias, variability, and statistics adaptation.

Theoretical Novelty: We refined Section 3 to clarify that applying the framework to the strictly bounded domain $ [0, 1] $ imposes unique mathematical constraints (Theorem 1), distinct from previous unbounded formulations (e.g., Hahn & Wei, 2024).

> Is the model comparison fair given the number of free parameters?

Yes. First, our primary metric Negative Log-Likelihood on held-out data inherently penalizes overfitting. Second, to address specific concerns about nonparametric flexibility, we computed a maximally conservative AICc upper bound, assuming the parameter count equals the full grid size. Even under this extreme penalty, the flexible Bayesian model outperforms all parametric baselines with the lowest Summed $\Delta$AICc (see captions of Appendix Figure 9, Figure 17, Figure 26).


> Are the results from the Adaptation task (N=26) robust?

To verify robustness, we performed a subject-level bootstrap analysis with 10,000 iterations. The flexible Bayesian model significantly outperformed competing accounts in $\ge 98.1\%$ of bootstrap samples (Appendix C10.3), confirming that the results are robust at the population level.

> Readers might need more intuition to better understand the model comparison.

We addressed this by adding Bias Decomposition analyses (Section 4.1 and Appendix C13.2) to visually disentangle the underlying mechanisms.

As a key example, the decomposition reveals that Uniform Encoding models lack Likelihood Repulsion. This forces them to rely exclusively on rigid boundary regression, intuitively explaining why they cannot capture the smooth S-shaped bias observed in human data.

This analysis also helps readers intuitively understand the trade-offs between different model components (e.g., how the Prior compensates for Encoding deficits) across different tasks.

---

### Meta-Review · Area_Chair_RgQE · 2026-01-11

**Summary:**

This paper proposes a Bayesian encoding–decoding framework to explain classic distortions in human probability perception, including inverse S-shaped probability weighting, variability, and adaptation to changing statistics. The core idea is that probability distortion emerges naturally from rational Bayesian inference over noisy internal representations of probability, grounded in principles of efficient coding. The authors develop a theoretical account characterizing properties of optimal encoding under bounded probability representations and empirically evaluate the framework across multiple domains, including probability estimation, economic pricing, risky choice, and a new adaptation experiment with bimodal priors. Across these tasks, the proposed Bayesian models are shown to outperform existing descriptive and bounded rationality models, including BLO and prospect-theoretic baselines.

Overall, reviewers viewed the paper as a technically careful and well-motivated attempt to provide a unifying normative perspective on probability weighting, with connections to Bayesian brain and efficient coding theories in cognitive science. The main strength lies in the theoretical framing, which aims to unify multiple strands of prior work under a single Bayesian account, together with empirical evaluations across several behavioral paradigms. The main concerns center on two dimensions: (1) the presentation, which makes it difficult for some reviewers to assess the conceptual and technical novelty of the contributions, and (2) the empirical evaluation, particularly whether comparisons sufficiently account for model complexity.

**Reviewer Concerns:**

Several reviewers raised concerns about whether the empirical advantages of the flexible Bayesian models, as well as the small sample size in the adaptation experiment, could be attributed to model capacity rather than substantive explanatory power. In response, the authors argue that held-out negative log likelihood guards against overfitting, provide a conservative AIC analysis assuming an upper-bound parameter count, and conduct a subject-level bootstrap analysis for the adaptation task to assess robustness under resampling. I find the AIC analysis helpful and believe it strengthens confidence that the reported gains are not trivially driven by unpenalized flexibility. However, the held-out likelihood and bootstrap analyses are limited in their ability to address broader questions about effective model complexity and generalization, especially in low-data regimes, and the bootstrap results apply only to the smallest dataset. Overall, I consider this concern to be moderately addressed by the AIC arguments, though it would benefit from more careful treatment.

The other shared concern relates to presentation, which limited the interpretability of both the conceptual and technical contributions. After reading the authors’ rebuttal and the revised manuscript, I tend to agree that the conceptual contribution is reasonable, and that providing a unified framework for examining the well-known probability weighting function is useful to the community. I do, however, continue to share reviewer KhJE’s concerns regarding the interpretation of the technical contributions, particularly with respect to their relationship to prior work by Hahn et al.

**Reviewer Scores:**

One reviewer was initially positive, with a score of 8.

Reviewer WQML gave an initial score of 4, with concerns mainly focused on presentation and clarity of contribution. Reviewer KhJE gave an initial score of 4, raising concerns about both presentation and the treatment of model complexity. Reviewer LsJ6 also gave an initial score of 4, with concerns primarily centered on the penalization of model complexity.

Based on the rebuttal and revisions, I would expect reviewer LsJ6 to increase the score to 6, though I am less confident about whether the other two reviewers would raise their scores.

Given that this is a borderline paper, I have read the paper myself. My own assessment aligns very closely with Reviewer KhJE’s evaluation. I find the paper to make an interesting and potentially valuable contribution, but the presentation still makes it difficult to assess the technical novelty with confidence. I view this as a borderline paper but might lean towards rejection. That said, I believe the paper has clear potential, and with more careful positioning relative to prior work and a clearer treatment of model complexity and generalization, it could become a strong future submission.

---

### Decision · Program_Chairs · 2026-01-26

Reject